UPDATE ARTICLE

# A chromosomal mutation is superior to a plasmid-encoded mutation for plasmid fitness cost compensation

**Rosanna C. T. Wright**[1], **A. Jamie Wood**[2,3], **Michael J. Bottery**[1], **Katie J. Muddiman**[1], **Steve Paterson**[4], **Ellie Harrison**[5], **Michael A. Brockhurst**[1], **James P. J. Hall**[4]*

**1** Division of Evolution, Infection and Genomic Sciences, University of Manchester, Manchester, United Kingdom, **2** Department of Biology, University of York, York, United Kingdom, **3** Department of Mathematics, University of York, York, United Kingdom, **4** Department of Evolution, Ecology and Behaviour, Institute of Infection, Veterinary and Ecological Sciences, University of Liverpool, Liverpool, United Kingdom, **5** School of Biosciences, University of Sheffield, Sheffield, United Kingdom

\* j.p.j.hall@liverpool.ac.uk

The Editors encourage authors to publish research updates to this article type. Please follow the link in the citation below to view any related articles.

## Abstract

Plasmids are important vectors of horizontal gene transfer in microbial communities but can impose a burden on the bacteria that carry them. Such plasmid fitness costs are thought to arise principally from conflicts between chromosomal- and plasmid-encoded molecular machineries, and thus can be ameliorated by compensatory mutations (CMs) that reduce or resolve the underlying causes. CMs can arise on plasmids (i.e., plaCM) or on chromosomes (i.e., chrCM), with contrasting predicted effects upon plasmid success and subsequent gene transfer because plaCM can also reduce fitness costs in plasmid recipients, whereas chrCM can potentially ameliorate multiple distinct plasmids. Here, we develop theory and a novel experimental system to directly compare the ecological effects of plaCM and chrCM that arose during evolution experiments between *Pseudomonas fluorescens* SBW25 and its sympatric mercury resistance megaplasmid pQBR57. We show that while plaCM was predicted to succeed under a broader range of parameters in mathematical models, chrCM dominated in our experiments, including conditions with numerous recipients, due to a more efficacious mechanism of compensation, and advantages arising from transmission of costly plasmids to competitors (plasmid "weaponisation"). We show analytically the presence of a mixed Rock-Paper-Scissors (RPS) regime for CMs, driven by trade-offs with horizontal transmission, that offers one possible explanation for the observed failure of plaCM to dominate even in competition against an uncompensated plasmid. Our results reveal broader implications of plasmid-bacterial evolution for plasmid ecology, demonstrating the importance of specific compensatory mutations for resistance gene spread. One consequence of the superiority of chrCM over plaCM is the likely emergence in microbial communities of compensated bacteria that can act as "hubs" for plasmid accumulation and dissemination.

**Data Availability Statement:** All data and analyses are provided on Zenodo https://doi.org/10.5281/zenodo.13963497 Raw data from flow cytometry experiments (relative fitness and dynamics experiments) are provided at https://doi.org/10.5285/51046841-deaa-422f-a303-2c0759f014b4, and processed data are on the University of Liverpool DataCat at https://doi.org/10.17638/datacat.liverpool.ac.uk/2585.

**Funding:** This work was supported by the Natural Environment Research Council (NE/R008825/1) to MAB, EH, AJW, SP, JPJH. RCW is supported by a funding from the Biotechnology and Biological Sciences Research Council to RCW and MAB (BB/T014342/1). JPJH is supported by a Medical Research Council Career Development Award (MR/W02666X/1). MJB is supported by the Wellcome Trust Sir Henry Wellcome Fellowship (221663/Z/20/Z). The funders had no role in study design, data collection and analysis, decision to publish, or preparation of the manuscript.

**Competing interests:** The authors have declared that no competing interests exist.

**Abbreviations:** CM, compensatory mutation; GLMM, generalised linear mixed effects model; LM, linear model; ODE, ordinary differential equation; RPS, Rock-Paper-Scissors.

# Background

Conjugative plasmids are important for bacterial evolution. Plasmids transfer niche-adaptive ecological functions between lineages and consequently can drive adaptation and genomic divergence [1–3]. However, acquiring a new conjugative plasmid is frequently costly for the host cell. Such plasmid fitness costs can arise from a variety of causes, including the metabolic burden of plasmid maintenance, disrupted gene regulation, stress responses, cytotoxicity, and mismatched codon usage [4]. The long-term persistence of costly plasmids within bacterial lineages can nonetheless be achieved by compensatory mutations (CMs), which negate these fitness costs [5]. Experimental evolutionary studies have revealed diverse CMs, which affect a wide range of gene functions, including regulatory genes, helicases, other co-resident mobile genetic elements, or hypothetical genes without known function [6–16].

The fitness cost of a given plasmid in a given host can be ameliorated by alternative CMs occurring in distinct genetic targets, sometimes encoded by different replicons, i.e., mutations of genes on the chromosome, which we term chrCM, or on the plasmid, which we term plaCM. This phenomenon is exemplified by the common soil bacterium *Pseudomonas fluorescens* SBW25 (henceforth "SBW25") and the environmental mercury resistance plasmid pQBR57. Both SBW25 and pQBR57 were isolated from sugar beet plants at a field site in Wytham Woods, Oxfordshire, United Kingdom in the 1990s [17,18]. pQBR57 causes a substantial fitness cost in SBW25 due to a specific genetic conflict with a chromosomal hypothetical gene PFLU4242, inducing a sustained SOS response and the maladaptive expression of chromosomal prophages leading to cell damage [12]. This costly cellular disruption can be negated by single CMs affecting either PFLU4242 itself or a plasmid-encoded regulator PQBR57_0059. Either CM is sufficient to reduce the fitness cost of pQBR57 and both fix the transcriptional disruption caused by pQBR57 acquisition. For clarity, we refer to SBW25 strains with a loss-of-function mutation in PFLU4242 as SBW25::chrCM to indicate chromosomal CM, and strains with a loss-of-function mutation in PQBR57_0059 as pQBR57::plaCM to indicate plasmid CM. Although both CMs evolved in SBW25 plasmid-carrying populations in potting soil microcosms, they were never observed to co-occur in the same genome, suggesting that there is no added benefit of combining both CMs in the same cell. ChrCM can ameliorate the fitness costs of other pQBR plasmids, whereas the benefits of plaCM are transmitted when pQBR57::plaCM transfers by conjugation [12,15]. Indeed, while chrCM was more commonly isolated from populations grown with selection (chrCM detected in clones from 75% of CM-containing populations), plaCM was the predominant CM in populations grown without selection in which the plasmid (initially) persisted by infectious transfer (plaCM detected in clones from 67% of CM-containing populations) [12].

Where alternative CMs exist on the chromosome or the plasmid, existing theory predicts that plaCMs will be superior [19,20]. This superiority arises because, unlike chrCMs which are only inherited vertically at cell division, plaCMs are also transmitted horizontally by conjugation. Provided the plaCM also negates the fitness cost of the plasmid in newly formed transconjugant cells, the linkage of the plasmid and the CM can thus enhance plasmid maintenance and spread. Correspondingly, plaCMs are predicted to outcompete chrCMs even if plaCMs offer less efficient amelioration than chrCMs. However, previous attempts to explore these predictions theoretically have relied on numerical simulations and extensive parameter fitting [19,20], limiting the generalisability of the findings, while experimental tests competing alternate modes of CM are lacking altogether. Moreover, measurements of diverse existing plasmids have suggested a direct mechanistic link between transmissibility and plasmid cost such that mutations to reduce cost may inevitably reduce transmission, while evolutionary

experiments have indeed reported a trade-off between evolved plaCMs and conjugation rate, potentially impeding the expected success of plaCMs [21–23].

To contrast the effect of chrCM with plaCM on bacteria-plasmid dynamics, we first develop 2 simple mathematical models each based on 3 coupled ordinary differential equations (ODEs) in which we consider the arrival of a plasmid-bearing strain with either a chrCM or a plaCM. A key strength of this approach is that the models we create can be solved exactly (in a dynamical systems sense) to provide general understanding without the need for parameter fitting [14,24]. The predictions generated by our models were then tested experimentally. To enable direct competition of chrCM and plaCM, we engineered variants of SBW25 and pQBR57 carrying defined CMs and fluorescent tags allowing cells containing the plaCM and/ or chrCM to be distinguished by flow cytometry. We then performed competition experiments across various ecological scenarios predicted to alter the differential benefits of these contrasting modes of compensatory evolution. Specifically, we varied the strength of mercury selection, the presence of other plasmid replicons in the population, or the availability of plasmid-free recipient cells for onward conjugative transfer within the population. We show that, contrary to expectations, plaCM performs poorly in competition with chrCM under all tested conditions due to lower efficacy of amelioration, a probable trade-off against horizontal transmission or establishment in transconjugants, and an overlooked benefit of chrCM that effectively enables these cells to "weaponise" costly plasmids to reduce the fitness of competitors. Our results have implications for the mobilisation of genes in microbial communities, suggesting that chromosomes are more likely to become plasmid-favourable—and thus hubs of source-sink horizontal gene transfer—than plasmids are to become low-cost generalists across hosts.

## Results

### Mathematical model of plasmid- or chromosome-encoded compensatory mutation dynamics

To understand the dynamics of chrCM and plaCM, we modified a simple, well-understood model of bacteria-plasmid population dynamics to include either chrCM or plaCM. Separate models were preferred because the combined system (i.e., containing both chrCM and plaCM) was too complex to solve analytically. Our goal was to gain a complete understanding of the resulting dynamical system—rather than a time-dependent solution—so that we have knowledge of the ultimate fate of a solution for arbitrary choices of parameters [14,24]. The basic model, without CMs, is detailed in Supplementary Text B in S1 File and recapitulated the key findings of previous studies wherein costly plasmids do not invade unless their conjugation rate, $\gamma$, is larger than $\mu(\alpha-\beta)/(\alpha-\mu)$, and only competitively displace the plasmid-free population if $\gamma$ is larger than $\mu(\alpha-\beta)/(\beta-\mu)$, where $\alpha$ is the plasmid free growth rate, $\beta$ is the plasmid containing growth rate, and $\mu$ is the population turnover rate. Positive selection for the plasmid is included via the selection pressure term, $\eta$, which was initially set to zero. Spontaneous plasmid loss by segregation was not incorporated, as previous experimental work showed that these events are rare on the timescale of the study such that population change is principally driven by competition with existing plasmid-free cells rather than their de novo generation by segregation [25], and including this term in the models complicates analytical progress.

We then considered how the addition of chrCM affects the outcome of this underlying basic system. Here, plasmid-free wild-type bacteria ($f$) and wild-type bacteria containing a wild-type plasmid ($p$) are invaded by a chrCM variant bearing a wild-type plasmid ($c$). The compensatory effect is assumed to be imperfect so the growth rate of the 3 strains are assumed to be $\alpha$ for $f$, $\beta_P$ for $p$, and $\beta_C$ for $c$, respectively, where $\alpha > \beta_C > \beta_P$. The dynamics of the system

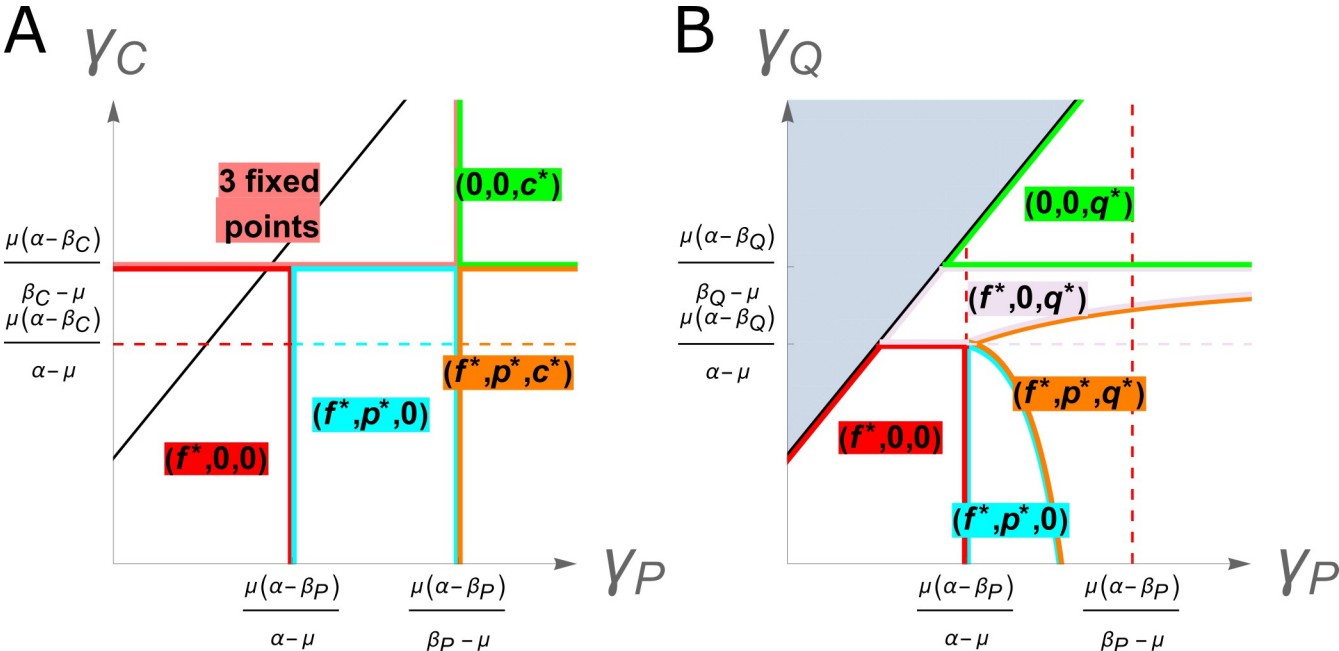

**Fig 1. Phase diagrams describing the fate of compensatory mutations.** (A) Chromosomal CMs. (B) Plasmid CMs. Axes describe relative conjugation rates without (x) and with (y) the corresponding CM, with the black line indicating no difference ($\gamma_P = \gamma_Q = \gamma_C$, note that plots focus on the region of interest and thus the black line does not intersect the point where the axes meet). Where $\gamma_Q > \gamma_P$ (i.e., both growth rate and conjugation rate are increased by the CM, indicated in grey) in the plasmid compensation model the wild-type plasmid is always lost, with the system reverting to the well-known two-member model of a plasmid-free and plasmid-containing type, as described in Supplementary Text B in S1 File. While similar behaviour occurs in the chrCM model, there is not the same mathematical simplification. Plots illustrating temporal dynamics for different regions of parameter space are shown in Fig A in S1 File. The data underlying this figure can be found in https://dx.doi.org/10.5281/zenodo.13963497.

are then described by the following set of ODEs:

$$\frac{df}{dt} = \alpha f(1 - f - p - c) - \mu f - \gamma_P p f - \gamma_C c f - \eta f, \tag{1}$$

$$\frac{dp}{dt} = \beta_P p(1 - f - p - c) - \mu p + \gamma_P p f + \gamma_C c f, \tag{2}$$

$$\frac{dc}{dt} = \beta_C c(1 - f - p - c) - \mu c. \tag{3}$$

In the case of no selection for the plasmid, $\eta = 0$, this set of ODEs can be solved exactly for the fixed points to yield a simple phase plane structure in which chrCM sweeps to fixation $(0, 0, c^*)$ after invasion by the mutated strain according to the value of the conjugation rate (Fig 1A, green region). When the underlying fixed point is plasmid-free only $(f^*, 0, 0)$ (i.e., $\gamma_P < \mu(\alpha - \beta_P)/(\alpha - \mu)$) or mixed $(f^*, p^*, 0)$ (i.e., $\mu(\alpha - \beta_P)/(\beta_P - \mu) > \gamma_P > \mu(\alpha - \beta_P)/(\alpha - \mu)$), and compensation is imperfect, the fixed points are separated by a saddle at $(f_s, p_s, c_s)$. This means that when the conjugation rate is sufficiently high the chrCM will always invade, but at lower conjugation rates (provided $\alpha > \beta_C$) there is a threshold. If the initial proportion of chrCM exceeds the threshold value (given by the saddle), complete replacement occurs and chrCM successfully invades and moves to fixation (Fig 1A, pink region). The reason invasion can occur despite chrCM having lower growth than the plasmid-free is that a plasmid-carrying chrCM can conjugate the costly plasmid into

plasmid-free competitors ("weaponisation"), such that chrCM then exceeds the uncompensated competitor's growth rate (i.e., $\beta_C > \beta_P$) and takes over the system. Selection ($\eta > 0$) reduces the relative ability of plasmid-free wild-type $f$ to compete against plasmid-bearers, with the result that the dynamics are driven by the competition between $p$ and $c$ and the difference between $\beta_C$ and $\beta_P$, thus favouring chrCM. This result is demonstrated in Supplementary Text B in S1 File.

We next considered a system invaded by a variant bearing a plaCM plasmid ($q$). This leads to a set of 3 differential equations analogous to those above:

$$\frac{df}{dt} = \alpha f(1 - f - p - q) - \mu f - \gamma_P p f - \gamma_Q q f - \eta f, \tag{4}$$

$$\frac{dp}{dt} = \beta_P p(1 - f - p - q) - \mu p + \gamma_P p f, \tag{5}$$

$$\frac{dq}{dt} = \beta_Q q(1 - f - p - q) - \mu q + \gamma_Q q f. \tag{6}$$

Analysis of the system reveals a more complex phase portrait, described in Fig 1B in the case of no selection, $\eta = 0$. Consistent with prior work [19], comparison of Fig 1A and 1B shows a wider range of conditions in which plaCM is successful compared with chrCM. Predominantly, this is where the CM enables plasmid invasion or dominance (purple and green regions in Fig 1B): biologically plausible conditions given previous observations of intermediate levels of plasmid carriage by populations [26]. Where there is no trade-off with conjugation rate, plaCM always displaces the uncompensated plasmid, invading the system if $\gamma_Q > \mu(\alpha - \beta_Q)/(\alpha - \mu)$ and dominating if $\gamma_Q > \mu(\alpha - \beta_Q)/(\beta_Q - \mu)$, and, unlike chrCM, in a manner that does not depend on initial conditions.

Previous experiments have shown that CMs can affect the ability of a plasmid to conjugate [21,22,27]. We therefore investigated how the success of each CM is affected by changes in the conjugation rate. For plaCM, if $\gamma_Q > \gamma_P$, i.e., plaCM confers a higher transfer rate than the wild-type plasmid, the wild-type plasmid is always lost from the system, and the outcome for plaCM collapses into the single-plasmid system described above (Supplementary Text B in S1 File). However, if there is a trade-off such that $\gamma_P > \gamma_Q$, various outcomes are possible depending on the other parameters, including loss of both plasmids ($f^*$,0,0), fixation of plaCM (0,0, $q^*$), coexistence between wild type and plasmid-free ($f^*$,$p^*$,0), and, unexpectedly, a state with a long-term stable coexistence between the $f$, $p$, and $q$ populations (Fig 1B, orange region), which would not be possible to find in a linearised adaptive dynamics approach. The stable fixed point is oscillatory in character (a stable spiral) and is driven by Rock-Paper-Scissors (RPS)-like nontransitive dynamics. When $f$ is large, this promotes the conjugative spread of the fastest conjugating population, $p$. When $p$ is large, $f$ is small, so the opportunities for conjugation are relatively low but the force of infection remains high due to high $\gamma_P$. Here, the $q$ outgrows the $p$. When $q$ is large opportunities for conjugation are also low but as $\gamma_Q$ is relatively low the $f$ can outgrow the $q$. This RPS-like dynamic is approximate as the interactions are not perfectly symmetric: in the prototypical model, each type has a direct impact on one other type and is directly impacted by the third. Here, the competition arises through a mixture of competitive growth and competitive infection and is different for each combination of subpopulations. The inequality $\gamma_P > \gamma_Q$ can be interpreted as applying on average, e.g., transconjugants could initially have an increased or reduced conjugative rate post conjugation before reverting to the long-term rate, and the ultimate fate of the system would be the same.

Conjugation rates can vary with chromosomal genotype over many orders of magnitude, even for the same plasmid [28], and there is evidence that chromosomally encoded genes can directly regulate conjugation gene expression [29–31], suggesting that chrCM could also result in changes to conjugation rate. We found that for chrCM, the system is generally robust to changes in conjugation rate provided compensation is sufficiently strong ($\beta_C > \frac{\mu(\gamma_C + \alpha)}{\mu + \gamma_C}$). In cases where chrCM has a more substantial negative effect on $\gamma_C$, the CM is lost (Fig 1A, red and blue regions), except where the conjugation rate from uncompensated strains is sufficiently high (Fig 1A, orange area, $\gamma_P > \frac{\mu(\alpha - \beta_P)}{\beta_P - \mu}$). Under these conditions, the force of infection of costly plasmids ensures that the frequency of plasmid-bearers in the system is maintained at a high enough level such that chrCM has sufficient competitive advantage to persist. Like the plaCM system, the chrCM system can also result in an oscillatory stable coexistent solution, driven by a similar form of RPS dynamics when $\gamma_P$ is large and $\gamma_C$ small, when $p$ invades $f$ populations, is displaced by $c$ mutants that are resistant to the costly plasmid, which are in turn susceptible to invasion by $f$ which can outgrow re-infection from $c$. A chrCM that increases conjugation rate is less expected biologically, but the overall long-term patterns would similarly remain dependent on the ability of the plasmid to dominate the system: if it can, then chrCM is always favoured owing to the fitness benefit it enjoys relative to uncompensated plasmid-bearers, if it cannot then the fate of chrCM is dependent on its initial frequency in the manner described above, with lower frequencies required for higher levels of $\gamma_C$ For both CMs, selection simplifies the dynamics by reducing the contribution of the plasmid-free population ($f$), concentrating the competition on the growth differences between uncompensated and compensated strains, and when selection is large the plasmid free population is simply eliminated.

Although four-equation models describing a direct competition between chrCM and plaCM cannot be solved analytically, some insight can be gained by comparing Eqs 3 and 6. Specifically, we can see that as the abundance of plasmid-free recipients decreases, the $\gamma_Q f$ component that positively affects the success of plaCM correspondingly decreases, such that the relative success of plaCM and chrCM is increasingly determined by the difference between $\beta_C$ and $\beta_Q$, i.e., the relative strengths of amelioration of the 2 CMs. Numerical calculations of the fixed points for experimentally measured parameter values likewise indicate that the extent of compensation is the overriding factor determining the relative success of competing CMs.

Overall, then, our models identify a broader range of parameter space in which plaCM is likely to succeed, relative to chrCM. However, we also predict that trade-offs between compensatory mutations and plasmid conjugation can have complex effects on the overall success of a CM, particularly in communities that contain a mixture of uncompensated plasmid-carrying and plasmid-free competitors. Selection reduces the complexity of the dynamics in both cases by removing potential recipients from the system, thus reducing the contribution of plasmid transmission to the dynamics, and increasing the dependency of the outcome on the relative strength of the CM. We note that these results are likely to represent biologically relevant parameter space, as our numerical models could generate results in all areas of the parameter phase diagrams from experimentally measured parameter values by manipulating conjugation rates over less than 2 orders of magnitude (Fig A and Table A in S1 File).

## Knock-outs of putative "cost" genes recapitulate compensatory mutations

To experimentally investigate the dynamics of plaCM versus chrCM, we established an experimental system in which fluorescently labelled strains were engineered with plaCM or chrCM, enabling enumeration by flow cytometry. To test that our newly engineered strains exhibited the fitness effects by flow cytometry that we have previously observed by CFU plating, we first

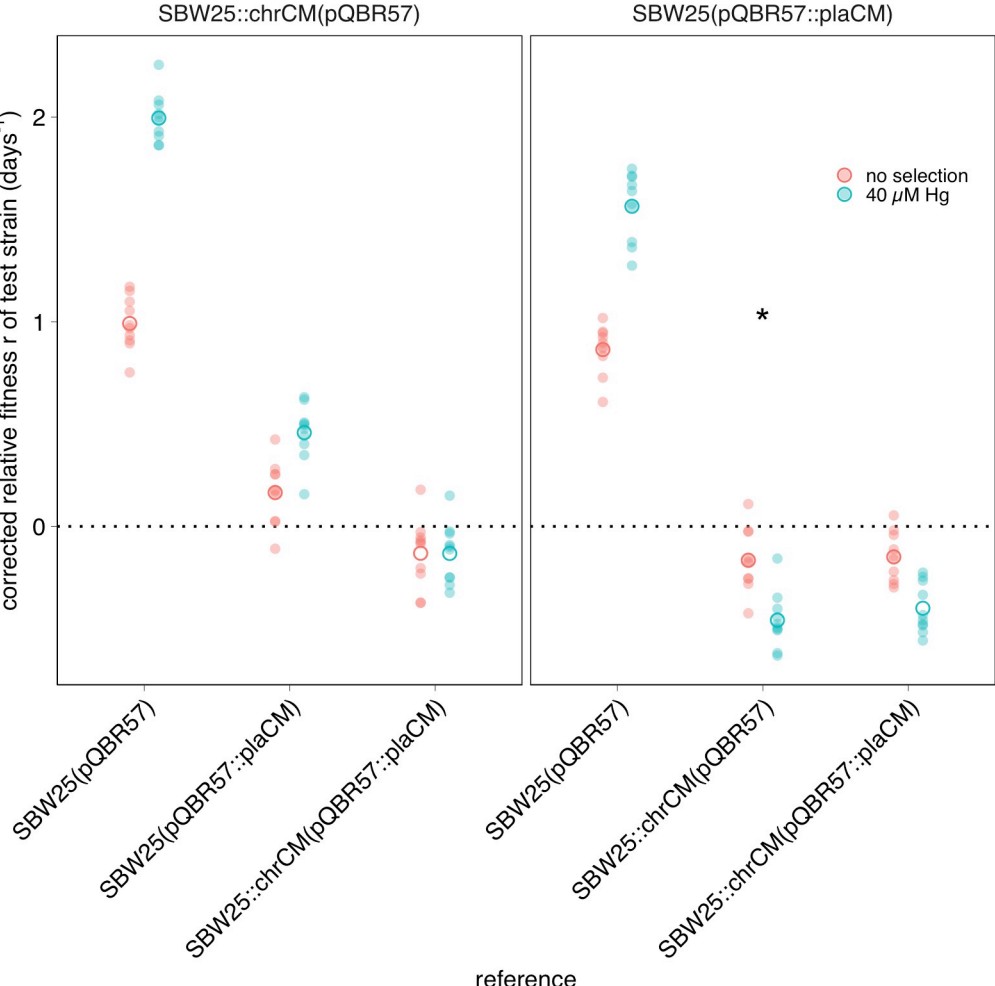

**Fig 2. Engineered plaCM and chrCM variants both ameliorate plasmid fitness costs, though chrCM is more effective.** Panels indicate test strains. Unfilled circles indicate mean across 10 replicates, each of which is indicated by a semi-transparent filled circle. The asterisk indicates data in the right panel which is also presented in the left panel. The data underlying this figure can be found in https://dx.doi.org/10.5281/zenodo.13963497.

performed 24-h competition experiments. As expected, each CM ameliorated the fitness cost of plasmid carriage (Fig 2, linear model [LM] one-sided post hoc comparison against 0: plaCM $t_9$ = 22.9, $p_{adj}$ < 1e-7; chrCM $t_9$ = 24.3, $p_{adj}$ < 1e-7). In addition, our head-to-head experimental design revealed that chrCM was fitter than plaCM (coefficient = 0.165, $t_9$ = 3.41, $p_{adj}$ = 0.015) and there was only a marginal fitness benefit of combining both CMs in the same cell (coefficient comparing chrCM versus double-compensation at no selection = −0.13, $t_9$ = −2.47, $p_{adj}$ = 0.07; coefficient = −0.13, $t_9$ = −2.86, $p_{adj}$ = 0.04 with selection). Interestingly, the benefits of CMs relative to non-compensated plasmid-carrying strains were enhanced in the presence of mercury, and more strongly so for chrCM than for plaCM (chrCM difference in coefficients = 1.0, plaCM difference in coefficients = 0.7), suggesting that the costly cellular disruption caused by pQBR57 is further exacerbated by exposure to mercury (as previously described by Carrilero and colleagues [32]). These results cause us to expect acceleration of CM invasion under mercury selection.

We previously did not observe any significant difference in the intraspecific conjugation rates of pQBR57 with or without chrCM or plaCM [12]. However, these experiments were

conducted over a relatively long time window (24 h), which could allow conjugation from transconjugants or fitness differences between transconjugants, donors, and recipients to mask differences in transfer rate [33,34]. We therefore re-measured conjugation rate of pQBR57 and pQBR57::plaCM using the Approximate Extended Simonsen approach [34]—an extension to the popular "Simonsen's gamma" [35] that accommodates the potential for variation in growth rate and secondary conjugation events. Again, we did not detect any significant difference in conjugation rate between the wild-type and plaCM variants of pQBR57 (Fig B in S1 File; $t_{9.86}$ = 0.75, $p$ = 0.94, TOST equivalence test with $\log_{10}$-transformed bounds ±0.5, $p$ = 0.02), and the similarity with previously measured values provided confidence that fitness differences and onward conjugation did not substantially affect our previous conjugation rate measurements [12]. Together, these measurements indicate that both CMs exhibit a fitness advantage over uncompensated pQBR57, with chrCM providing the greater benefit, and that neither CM exhibits a detectable trade-off with per capita conjugation rate.

## Chromosomal CM outcompetes plasmid-borne CM across various ecological conditions

To test our model predictions, we used our validated strains to test the relative success of chrCM versus plaCM under varying ecological conditions. First, we varied the strength of selection for the plasmid-encoded mercury resistance trait. Informed by our models, we predicted that selection would increase the success of chrCM relative to plaCM. This is because selection would remove plasmid-free recipients from the system, and as plasmids prevent superinfection by similar plasmids through surface exclusion and/or entry exclusion, plaCM would gain no benefit from its ability to transfer by conjugation. For these experiments, our strains were labelled to track the relative success of each mode of compensation, and so the label was inserted into the replicon encoding the compensatory mutation, i.e., the SBW25ΔPFLU4242 chromosome for chrCM and pQBR57ΔPQBR57_0059 for plaCM. SBW25::chrCM carrying wild-type pQBR57 was competed against pQBR57::plaCM in a wild-type chromosomal background in populations initially containing wild-type plasmid-free recipients at 50% frequency (1:1:2 chrCM:plaCM:recipients), and populations were transferred for 8 transfers (approximately 50 generations). Consistent with our predictions, mercury selection indeed favoured chrCM (Fig 3, generalised linear mixed effects model [GLMM] interaction effect of selection:transfer $\chi^2$ = 102.9, $p$ < 1e-7, Fig 3). However, chrCM was also fitter than plaCM without selection (coefficient for transfer = −0.32, z = −15.7, $p$ < 2e-16), suggesting that any benefits from the ability of the plaCM to transfer into the recipient pool were outweighed by the reduced amelioration provided by plaCM relative to chrCM (Fig 3).

In parallel, we investigated whether the benefits of chrCM could be increased in microbial communities hosting multiple costly plasmids. Our previous work showed that pQBR57 and pQBR103 could be harboured in the same cell. We also showed pQBR103 could be compensated by the chrCM, ΔPFLU4242, both by itself and together with pQBR57, whereas evidence suggested that plaCM exacerbated the cost of pQBR103 [12,32]. We therefore predicted that chrCM would be favoured over plaCM both with and without selection in pQBR103-harbouring communities, since chrCM would ameliorate both plasmids. We established experiments as with Fig 3 except replacing the plasmid-free subpopulation with a subpopulation carrying pQBR103. Indeed, in communities harbouring pQBR103, chrCM again outcompeted plaCM (Fig 4, GLMM coefficient for transfer = −0.30, z = −14.5, $p$ < 2e-16, Fig 4), but there was no detectable additional effect of pQBR103 on the relative success of chrCM versus plaCM.

We reasoned that increasing the pool of potential recipients may tip the balance towards plaCM, since conjugation (and thus transmission of the CM) could then play a bigger role in

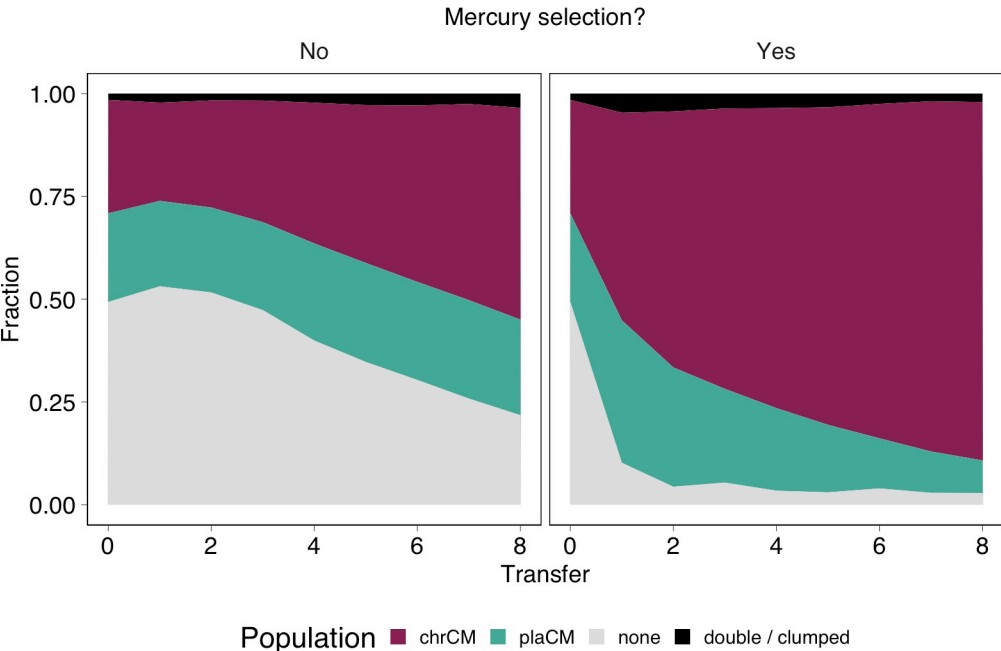

**Fig 3. Chromosomal compensatory mutations outperform plasmid-borne compensatory mutations, particularly under positive selection.** Subpanels indicate the presence of mercury selection (No/Yes). Mean of 20 replicates/ treatment (10 per marker orientation). Individual replicate plots are shown in Fig C in S1 File. The data underlying this figure can be found in https://dx.doi.org/10.5281/zenodo.13963497.

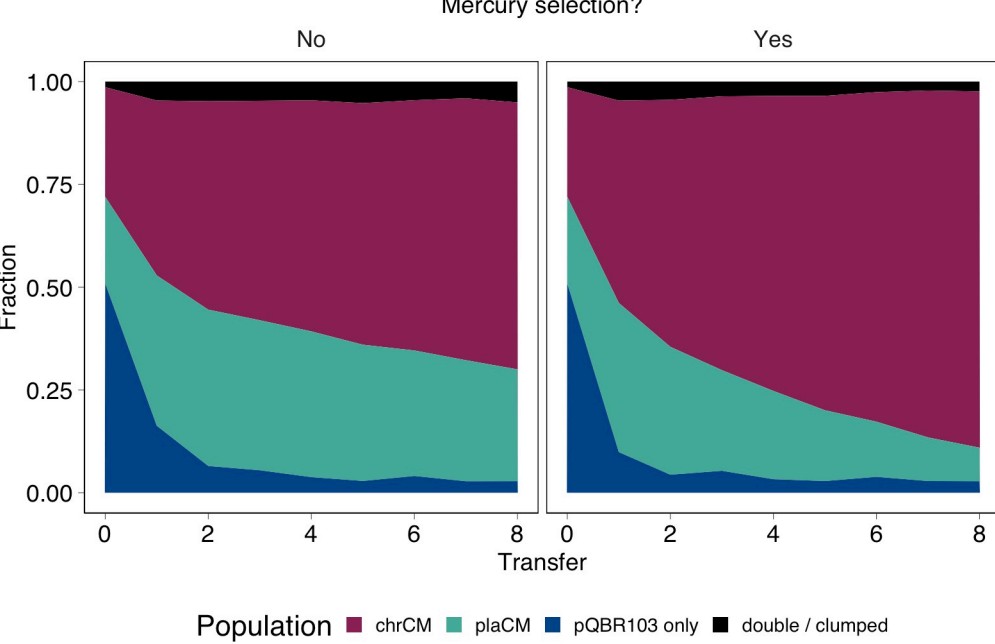

**Fig 4. Chromosomal compensatory mutations were not additionally favoured in environments with another costly plasmid.** Subpanels indicate the presence of mercury selection (No/Yes). Mean of 20 replicates/treatment (10 per marker orientation). Individual replicate plots are shown in Fig D in S1 File. The data underlying this figure can be found in https://dx.doi.org/10.5281/zenodo.13963497.

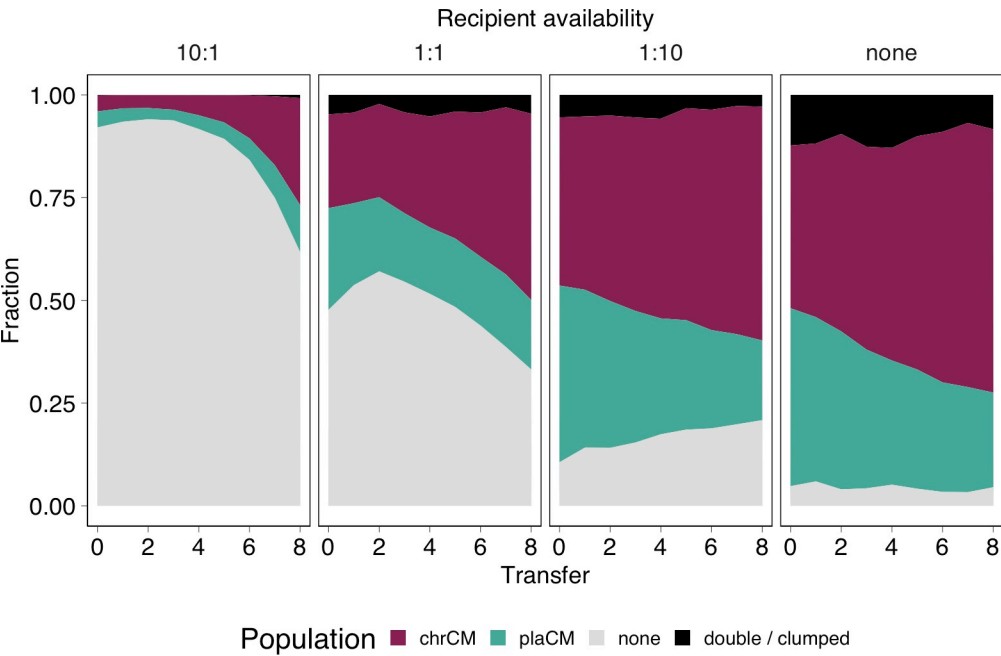

**Fig 5. ChrCM was more successful than plaCM regardless of recipient availability.** Subpanels indicate, from left to right, the starting fraction of wild-type/plasmid-free cells relative to a 50:50 mix of chromosomal CM (with wild-type plasmid) and plasmid CM in wild-type cells. Plots indicate the mean of 6 independent experiments; replicate-level plots are provided in Fig E in S1 File. The data underlying this figure can be found in https://dx.doi.org/10.5281/zenodo.13963497.

the population dynamics. We again established populations beginning with equal proportions of plaCM and chrCM but varied the proportions of recipients: (i) 10-fold excess of plasmid-free; (ii) equal abundance of plasmid-free; (iii) 10% plasmid-free; and (iv) without added plasmid-free. Given the lack of detected marker effect in Figs B and C (in S1 File), these experiments were conducted with SBW25::dTomato::chrCM(pQBR57) and SBW25(pQBR57::GFP::plaCM), and populations were propagated without selection. Contrary to expectations, plaCM performed relatively poorly against chrCM across all frequencies of plasmid-free recipients (Fig 5, GLMM transfer:ratio interaction $\chi^2$ = 5.00, $p$ = 0.17; main effect of transfer $\chi^2$ = 88.6, $p$ < 1e-7; coefficient for transfer = −0.35, z = −30.8, $p$ < 2e-16), suggesting that the potential benefits of CM transmission could not be manifested by the plaCM.

Our model predicted that transmission of wild-type plasmids from chrCM cells could play an important role in the success of chrCM. Furthermore, the theoretical advantage gained by plaCM through horizontal transfer may be reduced when in competition against wild-type plasmids carried by cells with chrCM, because wild-type plasmid transfer from chrCM could remove potential recipients for plaCM. However, our initial experiments used fluorescent labels to track the fates of the different compensatory alleles, i.e., the chromosomes of chrCM (SBW25::chrCM), and the plasmids of plaCM (pQBR57::plaCM), rather than the plasmids which began in each background. To understand how the plasmids themselves were affected by the different CMs, we established a complementary experiment to that in Fig 5, except, rather than tracking chrCM, we tracked the wild-type pQBR57 that began in the chrCM background (Fig 6), i.e., using SBW25::chrCM(pQBR57::tdTomato) and SBW25(pQBR57::plaCM:: GFP). Compared with pQBR57::plaCM, wild-type uncompensated pQBR57 from chrCM was significantly more successful, with a considerable proportion of the plasmids at the end of the experiment being wild-type uncompensated plasmids that began in the chrCM population,

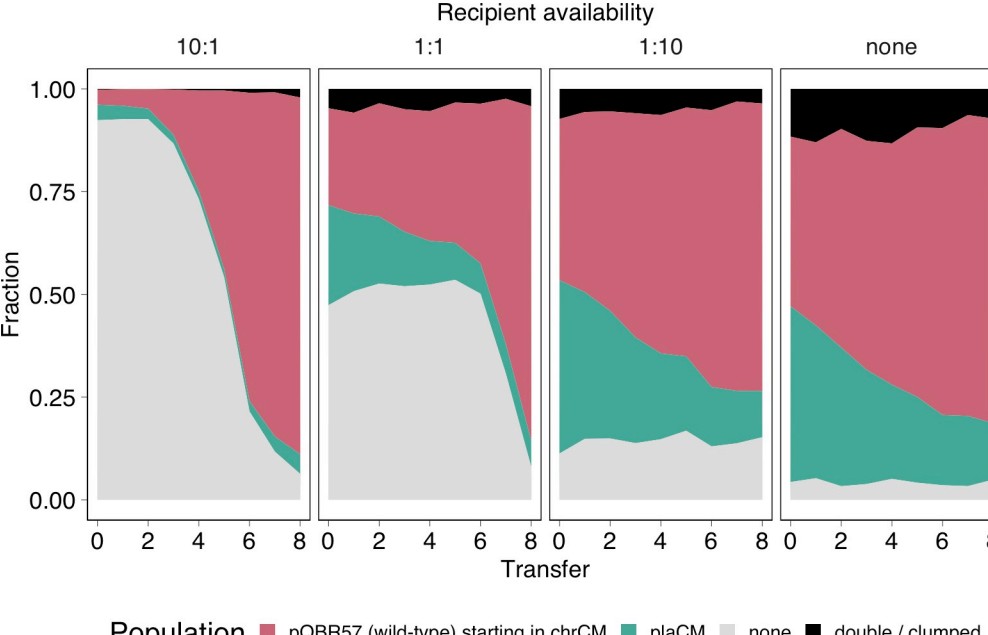

**Fig 6. Plasmids carried by chrCM cells were more successful than plaCM plasmids regardless of recipient availability.** Plots are arranged as Fig 5. Plots indicate the mean of 6 independent experiments; replicate-level plots are provided in Fig F in S1 File. The data underlying this figure can be found in https://dx.doi.org/10.5281/zenodo.13963497.

and patterns of invasion depending on the abundance of potential recipients (GLMM third-order polynomial transfer:ratio interaction $\chi^2$ = 287.4, $p$ < 1e-7). Notably, comparison with the experiments in Fig 5 (which were performed in parallel) revealed that invasion of the wild-type uncompensated plasmid from chrCM preempted the invasion of chrCM, indicating that plasmid transmission likely contributed to the overall success of chrCM. Essentially, as plasmid-free competitors are removed from the system by infection with the wild-type uncompensated plasmid, the space is filled by competition between CMs, which favours the more effective mechanism of compensation: chrCM. Indeed, when plasmid-free recipients were initially rare, the dynamics essentially come down to a competition between the compensatory mutations, with the most effective mechanism, chrCM, becoming dominant. To understand our experimental results further, we ran numerical simulations based on Eqs 1–6 and the equations in Supplementary Text A (in S1 File) parameterised with biologically plausible values (Table A in S1 File), which resulted in dynamics resembling those observed in the experiments (Fig G in S1 File). By contrast, inactivating the ability for plasmid "weaponisation" by chrCM (setting $\gamma_C$ = 0), fundamentally altered the dynamics, preventing chrCM fixation. The experimental results were therefore consistent with our model prediction that transmission of costly plasmids, and the consequent removal of plasmid-free recipients, facilitates the success of chrCM.

The relative inability of plaCM to invade the recipient population, when compared with wild-type pQBR57 from chrCM cells, suggested that either plaCM exerted a negative pleiotropic effect on the ability of pQBR57 to invade a recipient population, or, conversely, that chrCM enhanced the ability of wild-type pQBR57 to invade. To distinguish between these possibilities, we conducted a similar experiment to Fig 6, but here, plaCM was competed against pQBR57 harboured by an uncompensated competitor, i.e., SBW25(pQBR57::tdTomato) and

SBW25(pQBR57::plaCM::GFP). As with Fig 6, fluorescent labels were designed to track the relative success of the 2 plasmids. Unexpectedly, plaCM was only able to outcompete wild-type pQBR57 in the situation where plasmid-free recipients started at low frequency, i.e., effectively a head-to-head competition between compensated and uncompensated plasmid-bearers. We detected a significant effect of plasmid-free recipient ratio on the competition dynamics (GLMM fourth-order polynomial transfer:ratio interaction $\chi^2$ = 414.4, $p$ < 1e-7). Specifically, under conditions where recipients were more numerous, wild-type pQBR57 outcompeted the plaCM despite higher fitness costs (Fig 2), an observation that suggests that plaCM inhibits plasmid transmission or establishment in transconjugants relative to the wild type. Overall, plaCM did not appear to gain a substantive fitness benefit from horizontal transmission, only reaching dominance under conditions that prioritise vertical replication and only in populations without chrCMs.

Our analytical models for plaCM predicted cyclical RPS-like dynamics for some combinations of parameters (Fig 1B, orange region). The experimental results in Fig 7 were consistent with such nontransitive dynamics. Specifically, plasmid-free populations were invaded by the wild-type plasmid (left panel, and mid-left panel after transfer 4), the wild-type plasmid was outcompeted by plaCM (right panel, mid-right panel, and mid-left panel before transfer 4), and plaCM was outcompeted by plasmid-free (mid-right panel). To explore the dynamics in further detail, we compared experimentally determined parameters (Table A in S1 File) with the analytical model (Fig 1). With our system, and with experimentally measured parameter values, the uncompensated plasmid exceeds the threshold for plasmid invasion and domination, which is indeed what we have described previously [36]. Both chrCM and plaCM likewise exceed the threshold for domination by 18- and 10-fold, respectively. However, numerical simulations based on Eqs 1–6 and the equations in Supplementary Text A (in S1 File), and parameterised with biologically plausible values, required a considerable reduction of $\gamma_Q$ relative to $\gamma_P$ (10–100×) for the system to exist in the ($f^*,p^*,q^*$) region of parameter space that yields oscillatory RPS-like dynamics resembling the experimental results of Fig 7 (Fig I in S1 File)—a reduction inconsistent with the measurements of $\gamma_Q$ and $\gamma_P$ described above (Fig B in S1 File) or in our previous work. An interactive (Shiny) app enabling readers to explore these patterns is provided by following the link at www.jpjhall.net/plasmid-dynamics-model.

We therefore considered 2 other processes that could affect plaCM transmissibility beyond per capita conjugation rate: acquisition cost, and conjugation derepression. Each process has a transient effect on newly formed transconjugants. Acquisition costs are imposed by plasmids following arrival in a recipient, are often expressed through increased lag time, and are usually resolved in a matter of hours, in contrast to fitness costs which are genetically based, and inherited following cell division [37,38]. Conjugation derepression occurs when transconjugants exhibit a transient increase in onward conjugation rate, owing to a delay in the expression of the machinery that regulates conjugation genes [34,39]. If plaCM imposed greater acquisition costs, or a reduced degree of conjugation derepression, this might impede invasion of plaCM into a naïve population, relative to the wild-type pQBR57. To assess whether such effects could be measured experimentally in our system, we performed conjugation experiments with a 100-fold excess of recipients, such that conjugation from novel transconjugants would make an increased contribution to the calculated conjugation rate. However, no significant difference in conjugation rate was observed when comparing pQBR57 and pQBR57::plaCM (Fig J in S1 File, $t_{4.89}$ = 0.33, $p$ = 0.775). We also measured the lag time of de novo transconjugants upon gaining pQBR57 or pQBR57::plaCM, but again, we could not detect a trade-off (Fig K in S1 File), with de novo plaCM transconjugants in fact displaying a significantly reduced lag time compared with wild-type pQBR57 transconjugants ($t_{4.71}$ = 3.81, $p$ = 0.014). We therefore conclude that the differences in pQBR57 transmissibility between wild-type and

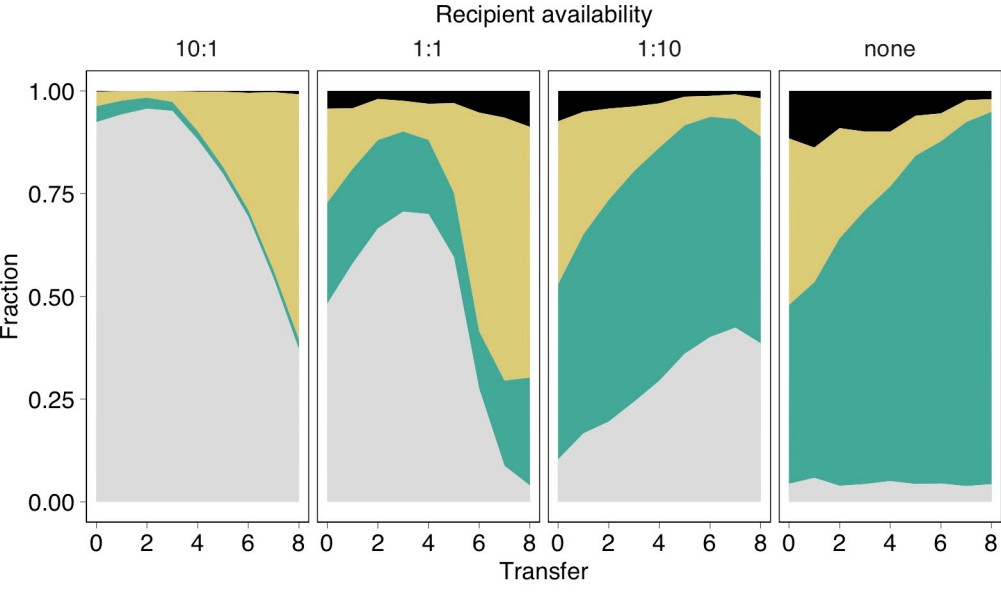

**Fig 7. Plasmid-free, uncompensated, and plaCM-compensated plasmids undergo "rock-paper-scissors" cyclical dynamics.** Plots are arranged as Fig 5. Plots indicate the mean of 6 independent experiments; replicate-level plots are provided in Fig H in S1 File. The data underlying this figure can be found in https://dx.doi.org/10.5281/zenodo.13963497.

plaCM variants predicted by our models and observed experimentally in Fig 7 emerge from yet-to-be-determined processes that will likely require further detailed examination of the molecular mechanisms underlying plaCM to elucidate.

## Discussion

Plasmid fitness cost amelioration is an important process driving the maintenance, distribution, and dissemination of plasmids and their associated accessory traits. Where plasmid fitness costs are generated from an interaction between the plasmid and resident chromosomal genes, mutations affecting either of these partners can enable plasmid survival. Previous experimental evolution studies on CMs have generally implicated chromosomal loci rather than plasmid loci as the principal targets, a finding which has been taken to reflect mutational supply, availability, and/or the poor ability of recessive compensatory mutations to penetrate when appearing on a multi-copy plasmid [11,20,40,41]. But although more difficult to access, plaCMs, once achieved, ought to be more successful than chrCMs under a range of ecological conditions owing to the simple fact that plaCM is propagated when the plasmid transfers into recipients [19]. The relatively high transfer rate of pQBR57 ought to have further accentuated this benefit [25], particularly under environmental conditions with high recipient availability. In contrast to these expectations, our analyses and experiments showed that plaCM was not successful under most tested conditions, losing out to chrCM, wild-type plasmids harboured by chrCM-containing cells, and even, where opportunities for HGT were plentiful, wild-type plasmids from cells lacking CMs.

There are several processes that could explain the relative failure of plaCM. First, our analyses of extensions to a simple plasmid population dynamics model reveal that for transmissible plasmids, chrCMs provide a hidden benefit besides directly reducing the fitness cost of

plasmid carriage to their bearers: conjugation from chrCM-containing cells transforms plasmid-free competitors into plasmid-carriers suffering the full burden of uncompensated plasmid carriage, indirectly enhancing the relative fitness of chrCM. Previous numerical simulations have likewise demonstrated the possibility for "weaponisation" of conjugative elements through compensatory mutation, and such a dynamic provides a further explanation for the persistence of non-beneficial plasmids in communities [20,42,43]. Our analyses generalise these findings and demonstrate that a high degree of amelioration and low impact on transmissibility will enhance chrCM invasion, particularly if the local chrCM frequency is sufficiently high, a condition that is more likely to be met in spatially structured habitats and/or where chrCM confers pleiotropic environmentally adaptive benefits [11,20,40]. Thus, the marginal benefits of conjugative transmissibility for plaCM in competition with chrCM are reduced.

Second, the benefits to plaCM of conjugative transmission wane as the plasmid-free recipient pool is diminished, from either (i) selection against plasmid-free recipients; or (ii) recipient acquisition of plasmids that can bar the incoming plaCM by incompatibility or exclusion. We observed both dynamics in our experiments. The effects of the latter mechanism are even more pronounced if there is a mechanistic trade-off between compensating plasmid fitness costs and the ability of the plasmid to transfer or establish in recipients, such as our results suggest, since a more transmissible (but more costly) plasmid can sweep through the recipient population, blocking access for the plaCM. Once the plasmid-free population is diminished, the dynamics of the system are driven by the relative growth rates of the different subpopulations, which in turn are determined by the degree of amelioration provided by chrCM and plaCM. In our system, SBW25::chrCM(pQBR57) outgrows SBW25(pQBR57::plaCM) in direct competition, because plaCM is not as efficient as chrCM at compensating the fitness cost of pQBR57. This discrepancy is likely due to the molecular mechanisms that underpin the principal fitness costs of pQBR57 and their resolution by compensatory mutation. PFLU4242 is a putative endonuclease, with a DUF262/DUF1524 domain structure resembling that of the GmrSD type IV restriction system [44], and so we hypothesised that this gene is somehow directly responsible for generating the dsDNA breaks that trigger the SOS response and subsequent toxic gene expression patterns characteristic of uncompensated pQBR57 carriage [12]. Loss-of-function mutations to PFLU4242 (i.e., chrCM) would directly prevent these breaks from occurring. On the other hand, *PQBR57_0059* encodes a lambda repressor-like protein that regulates expression of two other pQBR57-encoded putative DNA-binding proteins, PQBR57_0054–0055, and it is up-regulation of PQBR57_0054–0055 which provides the proximal mechanism of plaCM compensation, through a mechanism not yet fully understood. ChrCM therefore likely provides a more direct route than plaCM to resolving the genetic conflict at the heart of pQBR57-SBW25 fitness costs, and thus is mechanistically a more effective CM. Ultimately, it is likely to be the degree to which CMs reduce the cost of plasmid carriage, rather than CM transmissibility, which will primarily determine CM success.

Unexpectedly, plaCM was a poor competitor against the wild-type uncompensated pQBR57, winning out only in cases where there were few opportunities for conjugation. This experimental observation, coupled with our model parameterisation and numerical simulations, strongly implies that pQBR57::plaCM is not as effective as the wild-type plasmid at transmitting by conjugation and/or establishing in recipients. As measured per capita conjugation rates of wild-type pQBR57 and pQBR57::plaCM have been indistinguishable, even when controlling for the different fitness effects of the 2 plasmids or adjusting recipient proportions, these observations imply that differences in transmissibility emerge from processes other than plasmid transfer per se. PQBR57_0055 is a Spo0J/ParB-like protein, homologues of which have been shown to have various, nonspecific effects on gene expression [45,46], and by up-

regulating PQBR57_0054–0055, plaCM could have various and nuanced pleiotropic effects in transconjugants, but the underlying mechanisms resulting in its relative deficiency at horizontal transmission remain obscure. Variability in plasmid fitness effects have been observed at the level of single transconjugant lineages, as a plasmid establishes in a recipient, [37] and such effects are difficult to capture using the population-level approaches employed in this study. Nevertheless, our experiments comparing pQBR57::plaCM and wild-type pQBR57 are congruent with several other studies demonstrating trade-offs between vertical and horizontal plasmid transmission [6,22,23,27,47], and further show that imperfect plaCMs that trade-off against transmission can generate long-standing oscillatory dynamics which could sustain diversity in the plasmid population. The emergence of plaCM in our previous evolution experiments might be explained by benefits to plaCM in soil microcosms that were not recapitulated in broth culture [48]. Additionally, the accessibility of plaCM might be enhanced by the action of other MGEs in our system: in 7/9 cases disruption to *PQBR57_0059* was by transposon insertion, to which the plasmid, being relatively more AT-rich than the chromosome, could be more susceptible [25,49].

The chrCM in our system ameliorates diverse other mercury resistance plasmids, including pQBR103 and pQBR55 [12,15], and can ameliorate the costs of co-habiting compatible plasmids [32]. Previous work has likewise shown the generality of chromosomal compensatory mutations in reducing the fitness costs of different plasmids [11]. Our experiments did not detect a beneficial effect on chrCM versus plaCM when pQBR103 was introduced, likely because the low conjugation rate of pQBR103, overall benefit of chrCM, and short period of the experiment meant that any selective pressure imposed by pQBR103 acquisition was negligible. Nevertheless, the fact that chrCM was more likely to outcompete plaCM even in a single-plasmid (pQBR57) system indicates that lineages gaining CMs can become predisposed to acquiring further plasmids, potentially becoming hubs for horizontal gene transfer, plasmid recombination, and trait dissemination in microbial communities. Efforts to limit HGT, for example to control the spread of antibiotic resistance, should therefore focus on identifying and targeting such "keystone" strains. One possible route would be to use antagonistic parasitic MGEs such as lytic bacteriophage. The target of chrCM in our system (PFLU4242) appears related to GmrSD, a known genome defence mechanism [44]. Moreover, while little was known of the biological function of the *P. aeruginosa* PAO1 chrCM targets identified by San Millan and colleagues [13] at the time of discovery, gene function prediction tools now associate the accessory helicase PA1372 and partner gene PA1371 with genome defence ("Helicase + DUF2290 system") [50,51]. Likewise, the "Xpd/Rad3-like helicase" and "upstream UvrD helicase" targets of chrCMs identified by Loftie-Eaton and colleagues [11] in *Pseudomonas* sp. H2 refer to predicted components of prokaryotic Argonaute type III and Gabija, respectively. Similarly, in *Vibrio*, a recently discovered defence system DdmABC confers a high fitness cost on bearers of large plasmids such that they are removed from a population by purifying selection in a manner that resembles the large fitness cost imposed by PFLU4242 [52]. The ability of "MGE-favourable" organisms to receive and host plasmids thus likely trades off against susceptibility to costly parasitic elements, and exploiting this weakness may be a profitable approach to controlling the maintenance and spread of unwanted mobile genetic elements in various microbiomes.

Though some aspects—namely the relative degree to which chrCM and plaCM ameliorate plasmids and the extent to which plaCM imposes pleiotropic effects on plasmid transmission —may be system specific, the superiority of chromosomal CMs over plasmid-borne CMs in terms of mutational accessibility, indirect "weaponisation" effects on plasmid-free competitors (as presented here with a general theoretical mechanism), mechanistic efficacy, and reduced potential for trade-off with horizontal transmission, are likely to generalise to diverse other

plasmid-bacterial pairings, including pathogens and multidrug resistance plasmids. One broader implication is that transmissible plasmids thus have a limited ability to "act nice" by effectively ameliorating their own costs by evolution. Instead, plasmids are under stronger selection to improve their transmissibility and intracellular competitiveness, and it is largely down to resident genes to accommodate these unruly mobile genetic elements—or remove them.

## Methods

### Bacterial strains

Fluorescently labelled strains of *Pseudomonas fluorescens* SBW25 and *P. fluorescens* SBW25ΔPFLU4242 [15] were generated using the mini-Tn7 system and plasmid pUC18T-mini-Tn7T-Gm-eyfp, gifts from Herbert Schweizer via Addgene plasmids #65031 and #64968 [53] or a derivative in which dTomato was cloned to replace eyfp. The pQBR57ΔPQBR57_0059 knockout and fluorescently labelled variants of megaplasmid pQBR57 [18,25] were generated using homologous recombination with plasmid pTS-1 [54]. Briefly, for the knockout, 1 kb flanking regions of pQBR57 immediately upstream and down-stream of the PQBR57_0059 CDS (nucleotides 47925 and 48589 in the reference sequence LN713926) were amplified and cloned into the MCS of Xba-KpnI-digested pTS-1 using NEB HiFi assembly. For the fluorescently labelled plasmids, 1 kb fragments of pQBR57 targeting the site between PQBR57_0027–0028 at nucleotide 16050 in the reference sequence LN713926 and an expression cassette consisting of a Ptac promoter driving a fluorescence protein gene (either eGFP or tdTomato), followed by lambda t0 and rrnB T1 terminators were amplified and cloned into the MCS of XhoI/KpnI-digested pTS-1 using NEB HiFi assembly. Constructs were transformed into SBW25(pQBR57) by electroporation [53], and merodiploids selected on KB supplemented with 100 μg/ml tetracycline. Double-crossovers were selected on LB supplemented with 10% w/v sucrose and 20 μm $HgCl_2$, and candidates screened by PCR and tetracycline sensitivity before sending for whole genome sequencing ($2 \times 250$ bp, $>30\times$ coverage, MicrobesNG) to test for second-site mutations. Using breseq [55], no second-site mutations were detected for the strains used in these experiments. For experiments, each plasmid-containing replicate was established with an independent transconjugant from a genome-sequenced donor strain. Conjugation experiments were performed with non-fluorescent anti-biotically labelled strains described previously [12].

### Competition experiment

To determine the short-term fitness effects of compensation, direct competitions were performed between YFP- versus dTomato-labelled strains. For these experiments, all strains were chromosomally labelled. Overnight cultures were mixed at 1:1 ratio (test:reference) before inoculation at 1:100 dilution into 6 ml King's B media in a 30 ml glass universal with loose-fitting lid ("microcosm"), with or without mercury (Hg (II), 40 μm) and incubated at 28˚C, 180 rpm for 24 h. Each competition was repeated with 10 biological replicates. Flow cytometry was used to estimate bacterial counts: starting mixtures and endpoint competition cultures were diluted 1:100 into M9 buffer and run on a Beckman Coulter CytoflexS machine at 17 μl.min$^{-1}$ for either 5,000 counts (determined as events with signal in SSC-H channel $>10^3$) or 90 s maximum. Between samples, M9 buffer was sampled for 5 s to minimise cross-over between samples. Strain counts were determined by gating in the following channels: FITC-H (gate = $10^{3.4}$ for YFP) and PE-H (gate = $10^{3.4}$ for dTomato). Relative fitness was calculated as the difference

in Malthusian parameters, $r = \ln\left(\frac{test_{end}}{test_{start}}\right) - \ln\left(\frac{reference_{end}}{reference_{start}}\right)$. Control experiments to assess

potential marker effects, performed by competing isogenic, differently marked SBW25 (pQBR57) strains, showed a small, nonsignificant fitness cost of the marker (coefficient = $-0.03$, $t_9 = -0.77$, $p = 0.46$), and all values were corrected by subtracting the marker effect from the calculated relative fitness value. Other control experiments, performed to assess the effect of plasmid-encoded markers, produced results consistent with expectations from unmarked plasmids (Fig L in S1 File).

## Serial passage experiments

For all evolution experiments, bacterial populations were grown in 6 ml KB microcosms at 28°C with agitation at 180 rpm. Serial daily transfers of 1% population into fresh media were performed for 8 days, with daily flow cytometry used to track population dynamics. For flow cytometry, cultures were diluted 1:100 into M9 buffer and incubated with Hoechst 34580 stain (5 μg/ml) for 15 min in the dark at room temperature to enable detection of unlabelled bacterial strains. Flow cytometry data was sampled for 60 s at 17 μl.min$^{-1}$ with minimal gating for size (FSC-H $> 10^3$), and strain counts were determined in post analysis with the following thresholds: Hoechst-stained bacteria (i.e., total bacterial count, PB450-H $> 10^4$), YFP only (FITC-H $> 10^{3.5}$), dTomato only (PE-H $> 10^{3.4}$), YFP+dTomato clumps (FITC-H $> 10^{3.5}$ and PE-H $> 10^{3.4}$). Similar gates were used for GFP and tdTomato fluorescent proteins. Between samples, M9 buffer was sampled for 5 s to minimise cross-over between samples. For each fluorescently labelled strain, single-strain populations (3 biological replicates) were serially transferred for the duration of the experiment to ensure maintenance of the fluorescent signal.

To investigate the benefit of differing modes of compensation under varying selection pressures, we first challenged equal proportions of plaCM (SBW25(pQBR57Δ0059)) against chrCM (SBW25ΔPFLU4242(pQBR57)) in the presence of either a plasmid-free wild-type host (SBW25) or a wild-type host bearing a more costly conjugative plasmid (SBW25(pQBR103)). In all populations, the wild-type host started at approximately 50% frequency and carried a gentamicin resistance marker (Gm$^R$). In a fully factorial design, each population was grown in either in the presence or absence of mercury (Hg (II), 40 μm). Fluorescent markers (YFP or dTomato) associated with each mode of compensation allowed tracking of population dynamics in the presence of a non-fluorescent wild-type strain, with 10 biological replicates per marker orientation.

The impact of host availability on the benefit of plaCM was investigated in a separate evolution experiment, by challenging GFP-labelled plaCM (SBW25(pQBR57Δ0059::GFP)) against different host:plasmid backgrounds in the presence of varying ratios of plasmid-free wild-type hosts (SBW25::Gm$^R$), including 10× excess, equal proportions, 10× fewer and no available hosts. At each level of host availability, plaCM was competed against chrCM with a chromosomally encoded fluorescent label (SBW25Δ4242::dTomato(pQBR57)), chrCM with a plasmid-encoded fluorescent label (SBW25Δ4242(pQBR57::tdTomato)), or a wild-type plasmid bearer (SBW25(pQBR57::tdTomato)). For each treatment, 6 biological replicates were performed.

Raw data from flow cytometry experiments (relative fitness and dynamics experiments) are provided at https://doi.org/10.5285/51046841-deaa-422f-a303-2c0759f014b4, and the processed data used for figures and analysis are provided as part of the Zenodo repository associated with this article at https://dx.doi.org/10.5281/zenodo.13963497.

## Conjugation rates

Streptomycin-resistant *lacZ*-carrying donors (either SBW25::Sm$^R$(pQBR57) or SBW25:: Sm$^R$(pQBR57ΔpQBR57_0059)), Gm$^R$ recipients (SBW25), and Gm$^R$ transconjugants (either

SBW25(pQBR57) or SBW25(pQBR57ΔpQBR57_0059)), cultured overnight in 150 μl KB broth in an untreated CytoOne 96-well microtitre plate at 28˚C, were subcultured 1:30 into 150 μl fresh media and placed in a Tecan Nano plate reader for incubation at 28˚C with shaking. When exponential phase was reached (assessed by examination of growth curves; OD600 ~ 0.4), cultures were again diluted 30-fold into KB. Mixed cultures containing donors and recipients, or single-strain donor, recipient, or transconjugant cultures were sampled and cultured in the plate reader and spread on KB agar plates supplemented with 50 μg/ml X-gal for enumeration. After approximately 4 h growth, cultures were again sampled and spread on KB agar plates, some of which were supplemented with antibiotics (250 μg/ml streptomycin or 30 μg/ml gentamicin) and 20 μm mercury to enumerate transconjugants. Conjugation rates were calculated with the Approximate Extended Simonsen Method [34]. Excess-recipient conjugation experiments were performed as above, except using gentamicin-resistant donors (either SBW25(pQBR57) or SBW25(pQBR57ΔpQBR57_0059)) and streptomycin-resistant *lacZ*-carrying recipients (SBW25), and donor and recipient cultures were spread for enumeration before mixing equal volumes of recipient and a 1:100 dilution of donor and incubating in a CytoOne 96-well microtitre plate at 28˚C on a Stuart SSL-5 shaker for 18 h. Endpoint samples were spread for enumeration as above. While conjugation rate was therefore measured under slightly different conditions from those used in the serial passage experiment (glass vial versus 96-well polystyrene plate), consistency with previously measured rates in glass vials and the similarity between the 2 plasmid variants (indicated in Fig B in S1 File), suggest that such modifications do not significantly affect our conclusions. Data from these experiments are provided as part of the Zenodo repository associated with this article (https://dx.doi.org/10.5281/zenodo.13963497).

## Growth rates of fresh transconjugants

Conjugations were performed as described above, except using gentamicin-resistant donors (either SBW25(pQBR57) or SBW25(pQBR57ΔpQBR57_0059)) and streptomycin-resistant *lacZ*-carrying recipients (SBW25). Cultures were also set up using established transconjugants (SBW25::Sm$^R$(pQBR57) or SBW25::Sm$^R$(pQBR57ΔpQBR57_0059)) and grown into the exponential phase. After approximately 4 h growth of the cultures/co-cultures, strains were diluted 1:10 into KB in a 96-well plate supplemented with 40 μm HgCl$_2$ and 200 μg/ml streptomycin and incubated shaking for 1 h. Samples were then pelleted at 1,000 $g$ for 10 min, washed with 1 ml KB, and resuspended in 200 μl KB. Samples were diluted and spread on KB agar for enumeration, and growth curves were set up in KB supplemented with 200 μg/ml streptomycin and 20 μm HgCl$_2$ in the Tecan Nano plate reader for incubation at 28˚C with shaking and measurements (OD600) every 15 min. Data are provided as part of the Zenodo repository associated with this article (https://dx.doi.org/10.5281/zenodo.13963497).

## Statistics

Relative fitness was analysed using linear models, with post hoc pairwise comparisons performed using the package emmeans [56] and Bonferroni-corrected $t$ tests. Dynamics were analysed with GLMMs using the R package glmmTMB [57], with a beta-binomial response distribution, a logit link function, and the counts of each competitor as the response variables. Preliminary analyses identified overdispersion in the data, justifying the use of a beta-binomial rather than a binomial response distribution [58]. Non-independence of measurements arising from repeated sampling of populations was accommodated with random effects of "population" on intercept and slope, except for the experiment presented in Fig 7 which included only the random effect on intercept due to extremely low variance and high correlation between

random effects preventing model convergence. For experiments presented in Figs 6 and 7, polynomial terms were added to accommodate potentially non-monotonic relationships between competitors over time. Significance of fixed effects was determined by comparison of nested models using likelihood ratio tests. Conjugation rate data were analysed by *t* test and 2 one-sided tests using the package TOSTER [59]. Growth curve data were analysed using gcplyr [60]. Analysis scripts are provided on GitHub as part of the Zenodo repository associated with this article (https://dx.doi.org/10.5281/zenodo.13963497).

## Supporting information

**S1 File. Supplementary Information. Fig A.** Example dynamics from different regions of parameter space shown in Fig 1. Numerical simulations of a continuous flow model using the following parameters: $\alpha$ = 0.5 h$^{-1}$, uncompensated plasmid-bearer relative fitness = 0.82, compensated plasmid relative fitness = 0.95, $K$ = 5.7 × 10$^9$ ml$^{-1}$, $\mu$ = 0.04125. The base conjugation rate $\gamma$ was set above the "domination" threshold $\gamma_{dom} = \mu(\alpha - \beta)/(\beta - \mu)$ at 9.9 × 10$^{-12}$ ml. cells$^{-1}$h$^{-1}$ and was reduced below $\gamma_{dom}$ or the "invasion" threshold $\gamma_{inv} = \mu(\alpha - \beta)/(\alpha - \mu)$ for compensated and uncompensated plasmids-bearers to explore parameter space. The compensated plasmid conjugation rate $\gamma_C = \gamma_Q$ was set at $\frac{\gamma}{2}$ for cases where $\gamma_C = \gamma_Q > \gamma_{dom}$, at $\frac{\gamma}{10.5}$ where $\gamma_{inv} < \gamma_Q = \gamma_C < \gamma_{dom}$, and at $\frac{\gamma}{12}$ for cases where $\gamma_Q = \gamma_C < \gamma_{inv}$. The uncompensated plasmid conjugation $\gamma_P$ rate was set at $\frac{\gamma}{3.1}$ for $\gamma_{inv} < \gamma_P < \gamma_{dom}$ and at $\frac{\gamma}{3.5}$ where $\gamma_P < \gamma_{inv}$. Details on parameterisation are provided in Table A in S1 File. (A) Examples across parameter space for chrCM with initial density at 0.01 $K$, initial plasmid carriage at frequency of 50%, with 50% of these with CM. Coloured bars below each subpanel header indicate the corresponding regions of parameter space highlighted in Fig 1. (B) As (A) but for plaCM. (C) Different initial conditions for $\gamma_{inv} < \gamma_P < \gamma_{dom}, \gamma_{dom} < \gamma_C$ (the pink region of Fig 1A). In both cases, plasmid-bearers start at 50% of the total, but in the left hand-panel the CM is at 50% of this population, whereas on the right it is 0.01%. The boundary for the initial conditions, solved analytically, is provided in Supplementary Text B in S1 File, and the data underlying this figure can be found in https://dx.doi.org/10.5281/zenodo.13963497. **Fig B.** Approximate Extended Simonsen conjugation rates for pQBR57 and pQBR57::plaCM. The dotted line indicates the previously measured conjugation rate for wild-type pQBR57 [12]. The data underlying this figure can be found in https://dx.doi.org/10.5281/zenodo.13963497. **Fig C.** Individual replicates for the summarized data presented in Fig 3. Red/Yellow and Yellow/Red refer to the orientation of the fluorescent markers (chrCM / plaCM), and Yes / No refers to mercury selection. The data underlying this figure can be found in https://dx.doi.org/10.5281/zenodo.13963497. **Fig D.** Individual replicates for the summarized data presented in Fig 4. Red/Yellow and Yellow/Red refer to the orientation of the fluorescent markers (chrCM / plaCM), and Yes / No refers to mercury selection. The data underlying this figure can be found in https://dx.doi.org/10.5281/zenodo.13963497. **Fig E.** Individual replicates for the summarised data presented in Fig 5. The data underlying this figure can be found in https://dx.doi.org/10.5281/zenodo.13963497. **Fig F.** Individual replicates for the summarised data presented in Fig 6. The data underlying this figure can be found in https://dx.doi.org/10.5281/zenodo.13963497. **Fig G.** (A) ODE-based model simulations resemble plaCM versus chrCM experimental results in main text Figs 3, 5, and 6. Numerical simulations of a continuous flow model based on the following parameters: $\alpha$ = 0.54 h$^{-1}$, SBW25::chrCM(pQBR57) relative fitness = 0.97, SBW25(pQBR57::plaCM) relative fitness = 0.95, $K$ = 5.7 × 10$^9$ ml$^{-1}$, chrCM conjugation rate $\gamma_C$ = 4.6 × 10$^{-12}$ ml.cells$^{-1}$h$^{-1}$, plaCM conjugation rate $\gamma_Q = \frac{\gamma_C}{100}$ (see main text and Fig I in S1 File for discussion). Details on parameterisation are provided in Table A in S1 File. (B) Plasmid weaponisation can drive outcomes. As panel A, except chrCM conjugation rate $\gamma_C$ was set to zero. The qualitatively

different outcomes between panels A and B demonstrate the role that costly plasmid transmission from compensated strains can play. An interactive version of this figure is provided by following the link at www.jpjhall.net/plasmid-dynamics-model. The code underlying this figure can be found in https://dx.doi.org/10.5281/zenodo.13963497. **Fig H.** Individual replicates for the summarised data presented in Fig 7. The data underlying this figure can be found in https://dx.doi.org/10.5281/zenodo.13963497. **Fig I.** ODE-based model simulations resemble experimental results for low $\frac{\gamma_Q}{\gamma_P}$ ratios. Numerical simulations of a continuous flow model based on the following parameters: $\alpha$ = 0.54 h$^{-1}$, SBW25(pQBR57) relative fitness = 0.82, SBW25 (pQBR57::plaCM) relative fitness = 0.95, $K$ = 5.7 × 10$^9$ ml$^{-1}$, uncompensated conjugation rate $\gamma_P$ = 4.6 × 10$^{-12}$ ml.cells$^{-1}$h$^{-1}$, plaCM conjugation rate $\gamma_Q = \frac{\gamma_P}{100}$. Top panels indicate short-term dynamics, bottom panels longer-term dynamics. Details on parameterisation are provided in Table A in S1 File. An interactive version of this figure is provided by following the link at www.jpjhall.net/plasmid-dynamics-model. The code underlying this figure can be found in https://dx.doi.org/10.5281/zenodo.13963497. **Fig J.** An excess of recipients does not favour wild-type pQBR57. Conjugation experiments were established with a 100× excess of recipients and run for 18 h. The dotted line indicates the previously measured conjugation rate for wild-type pQBR57 [12]. The data underlying this figure can be found in https://dx.doi.org/10.5281/zenodo.13963497. **Fig K.** De novo plaCM transconjugants do not suffer increased lag time. Growth curves were conducted with fresh transconjugants under selection. PlaCM had a significantly reduced lag time compared with wild-type pQBR57 (t$_{4.71}$ = 3.81, $p$ = 0.014), but only in the de novo transconjugants. Time is in hours. The data underlying this figure can be found in https://dx.doi.org/10.5281/zenodo.13963497. **Fig L.** Control experiments testing fluorescently labelled strains produced results consistent with expectations. Unfilled circles indicate the mean across 10 (where test = SBW25::YFP(pQBR57)) or 8 replicates. Competitions between differently labelled isogenic-labelled strains all produced nonsignificant results ($p$ > 0.2) and small coefficients (<0.061) by $t$ test when comparing with $r$ = 0, and uncompensated plasmid fitness costs were recapitulated in competition against wild-type (SBW25::YFP) and plaCM ($p$ < 0.00002 in both cases, coefficients 0.68, and 0.54, respectively). A small (coefficient 0.12), marginally significant cost of plaCM ($p$ = 0.021) was detected in competition with plasmid-free (SBW25::RFP), consistent with other observations that plaCM amelioration may be less complete than alternative CMs. The data underlying this figure can be found in https://dx.doi.org/10.5281/zenodo.13963497. **Table A.** Experimentally measured parameters used to inform numerical simulations. **Text A.** Details of numerical simulation model. **Text B.** Model analyses.
(PDF)

## Author Contributions

**Conceptualization:** Rosanna C. T. Wright, A. Jamie Wood, Katie J. Muddiman, Steve Paterson, Ellie Harrison, Michael A. Brockhurst, James P. J. Hall.

**Data curation:** Rosanna C. T. Wright, James P. J. Hall.

**Formal analysis:** Rosanna C. T. Wright, A. Jamie Wood, Michael J. Bottery, James P. J. Hall.

**Funding acquisition:** A. Jamie Wood, Steve Paterson, Ellie Harrison, Michael A. Brockhurst, James P. J. Hall.

**Investigation:** Rosanna C. T. Wright, A. Jamie Wood, Michael J. Bottery, James P. J. Hall.

**Resources:** Rosanna C. T. Wright, Katie J. Muddiman, James P. J. Hall.

**Visualization:** Rosanna C. T. Wright, A. Jamie Wood, Michael J. Bottery, James P. J. Hall.

**Writing – original draft:** Rosanna C. T. Wright, A. Jamie Wood, Michael A. Brockhurst, James P. J. Hall.

**Writing – review & editing:** Rosanna C. T. Wright, A. Jamie Wood, Michael J. Bottery, Katie J. Muddiman, Steve Paterson, Ellie Harrison, Michael A. Brockhurst, James P. J. Hall.

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
