## [Editor Report · Decision Letter 0]

26 Jan 2024

Dear Jamie, 

Thank you for submitting your manuscript entitled "Superiority of chromosomal compared to plasmid-encoded compensatory mutations" for consideration as a Update Article by PLOS Biology.

Your manuscript has now been evaluated by the PLOS Biology editorial staff, as well as by an academic editor with relevant expertise, and I'm writing to let you know that we would like to send your submission out for external peer review.

Once your full submission is complete, your paper will undergo a series of checks in preparation for peer review. After your manuscript has passed the checks it will be sent out for review. To provide the metadata for your submission, please Login to Editorial Manager (https://www.editorialmanager.com/pbiology) within two working days, i.e. by Jan 30 2024 11:59PM.

Kind regards,

Roli

Roland Roberts, PhD

Senior Editor

PLOS Biology

rroberts@plos.org

---

## [Decision Letter · Decision Letter 1]

16 Apr 2024

Dear Jamie,

Thank you for your patience while your manuscript "Superiority of chromosomal compared to plasmid-encoded compensatory mutations" was peer-reviewed at PLOS Biology. It has now been evaluated by the PLOS Biology editors, an Academic Editor with relevant expertise, and by three independent reviewers. 

You'll see that reviewer #1 is mostly positive, but was disappointed to find out at the end that conjugation rates can’t explain differential compensatory mutation success, and that the explanation must lie elsewhere; he suggests either exploring another possible cause (e.g. acquisition costs), or rearranging the narrative to avoid disappointment. Reviewer #2 finds the paper interesting, but wants you to tone down the Title and discuss a few points (including acquisition costs) – s/he suggests some minor experiments too. Reviewer #3 is also fairly positive, but questions your parameter ranges, thinks there’s a disconnect between your modelling and what the experiments show, and wants you to make the modelling section more accessible to the general reader; as a mathematician, s/he find some aspects of the model confusing, and suggests several ways in which you could make more use of it to explore your system.

IMPORTANT: I discussed these comments with the Academic Editor, who agreed that we should invite you to address all of the reviewers' concerns. They especially asked me to flag the following for particular attention:

(i) Reviewer #1's suggestions to restructure the presentation of the results (including earlier presentation of the measured conjugation rates) and to test at least for increased acquisition/establishment costs of plaCM (using the protocol by Lopatkin).

(ii) Reviewer #3's suggestions to improve accessibility of the modelling.

(iii) In response to all three reviewers, to tone down the suggested generality of your findings, which are based on only one chrCM and one plaCM for one specific plasmid/host combination. This means adapting the title ("Superiority of a chromosomal to a plasmid-encoded compensatory mutation"), abstract and discussion.

In light of the reviews, which you will find at the end of this email, we would like to invite you to revise the work to thoroughly address the reviewers' reports.

Given the extent of revision needed, we cannot make a decision about publication until we have seen the revised manuscript and your response to the reviewers' comments. Your revised manuscript is likely to be sent for further evaluation by all or a subset of the reviewers.

**IMPORTANT - SUBMITTING YOUR REVISION**

*Re-submission Checklist*

*Published Peer Review*

*PLOS Data Policy*

*Blot and Gel Data Policy*

Sincerely,

Roli

Roland Roberts, PhD

Senior Editor

PLOS Biology

rroberts@plos.org

REVIEWERS' COMMENTS:

Reviewer #1:

[identifies himself as Alvaro San Millan]

In this study, Rosanna C.T. Wright and collaborators compare the ecological effects of chromosomal-encoded (chrCM) and plasmid-encoded (plaCM) compensatory mutations reducing the fitness costs associated with plasmids. The authors combine theoretical and experimental approaches to test the outcomes of the different CM in bacterial populations. They report that, in contrast to previous results in the field, chrCM are generally superior under a wide range of conditions. In general, I think that this is a very interesting paper and it combines modeling and experimental results in a very elegant way. The experimental system, which has been previously characterized by the authors, is now improved by using fluorescent tags and flow cytometry, which is a great addition. The results are novel, and uncover important ecological dynamics previously overlooked. The paper is beautifully written and the figures are clear and informative. I have however a major concern:

The two main factors affecting the success of the different CM (plasmid or chromosome) are (i) the degree of compensation associated with the CM, and (ii) the rate of conjugation of the plasmid. The authors showed that chrCM alleviated the cost better than chrCM, but although the conjugation rate is the other key parameter, they proceed with the theoretical model and all the subsequent experiments without experimentally measuring it. Then, all the results suggest that one of the most important reasons for the plasmid carrying the CM not to increase in frequency in the population has to be that this mutation reduced the conjugation rate of the plasmid. Up to that point all the results make complete sense (including very elegant and robust experiments). But at the very end of the results section, the authors measure the conjugation rate of each version of the plasmid, and discover that they are basically identical (while the model predicts a big difference according to the experimental results: 9.9 x 10-12 vs 1.3 x 10-13 ml.cells-1h-1). The authors suggest that "yet-to-be determined processes following plasmid establishment in recipient cells" should probably be responsible for the results, given that there is no difference in conjugation rate, and they are probably right, but they do not test them. The processes suggested by the authors in the discussion are: (i) a higher acquisition costs of plaCM compared to the wt plasmid or (ii) differences in the duration for conjugation derepression. Measuring acquisition costs is something that can be done relatively easily (as in Allison Lopatkin papers) maybe measure those? The relative relevance of secondary conjugations is more difficult to determine, but could also be done (by using recipient bacteria unbale to further propagate the plasmid, we do that using CRISPRi) or by performing mating assays with different frequencies of primary donors that may potentiate the effects of secondary conjugations (but I see how this can be a lot of work and maybe beyond the scope of the paper). I think that the work would be much more robust if the authors try to find the reason why the compensated plasmid is less able to spread. 

I guess that what I found a little disappointing as a reader, is that the entire paper builds around the results being explained by the difference in conjugation rate, which is finally not the case. Maybe a different structure (presenting experimentally determined conjugation rates at the beginning of the paper?) could change that feeling. In fact, the paper would still be very strong, because there are other important results and ecological dynamics described here. 

Minor comments 

Figure 1. "a" and "b" are missing in the panels. 

Figure 3 and 4 could be 2 panels of the same figure, and figure 5, 6 and 7 could also be panels of the same figure, in case there is a limited number of figures.

Figure 5: maybe indicate in the legend that there is no Hg selection.

Figure S6 Are the conjugation done under the same experimental conditions as the transfer experiments? In the methods it seems as if they could be a bit different (maybe this affects the rates?)

Alvaro San Millan

Reviewer #2:

In this manuscript, the authors perform both models and experiments comparing plasmid- and host-encoded mutations compensating for plasmid cost in P. fluorescens SBW25 and conjugative plasmid pQBR57. 

Contrary to model expectations, the experiments performed show that chromosomal CMs fare better than plasmid CMs, even in the presence of plasmid-free recipients. This is due to chromosomal CMs having reduced costs compared to plasmid ones, in competition in the absence of recipients, but more surprisingly also in the context of conjugation. Despite this, plasmid CMs do not have reduced conjugation rate compared to wt plasmids, as measured by classical assays. 

My main comment is that the conclusions (especially the title) should be nuanced: the conclusions of the modelling are that a wider range of conditions can promote plasmid CMs vs chromosomal ones. This is tested experimentally with only one couple of mutations, and specifically only one plasmid CM. The authors find a higher benefit of the chromosomal CM to vertical transmission; and an unexpected cost of the plasmid CM to horizontal transmission. That is fascinating and investigated in detail here, but I don't think this unique example can lead to a plural in the title of the article - the generality of the findings depends on different plasmid CMs having the same consequences on transmission, this is not obvious and not shown here. 

It might also be worth discussing more how these results can explain observed experimental evolution results: in the previous paper(s) with this model system, were the chromosomal CMs obtained more frequently than the plasmid ones, in which conditions, did they invade faster? 

Minor comments: 

L61 CMs "affecting" distinct genetic targets is confusing, as they can have (fitness) effects on both - replace with 'localised on' or similar to make it clearer? 

L91-92 and later: what the authors mean could be made more explicit to place the cited papers in the context of CMs evolution - these papers do not directly deal with CMs 

Did the authors perform any measure of fluorescent marker cost by competing isogenic strains differing only in the markers? 

Could the authors also discuss the amplitude of fitness effects compared to previous results? Selection coefficients ~ 1 in the absence of selection seem quite large. I had a rapid look at the previous PLoS Biology paper (Fig 2) where the amplitude seems lower but can't easily compare relative fitness W and selection coefficient results. 

Not strictly required for the conclusions here, but if strains with the right markers are available, it would be interesting to see an experiment similar to Fig 6 where pQBR57 wt within the uncompensated ancestor competes with the wt plasmid in chrCM - that would show nicely if higher fitness in the donor leads to higher transmission overall. 

L535 as the authors showed, the pleiotropic effects are not strictly on conjugation rate itself

Finally, maybe the two cited papers on acquisition cost could be discussed in a bit more detail. They do include modelling showing the effects of acquisition cost on conjugation dynamics, and showing that an intermediate acquisition cost can benefit a plasmid more than a low acquisition cost, which might be relevant to discuss here (I'm not sure how conceptually equivalent this is, but that appears at least similar to the 'paradox' of plasmid CMs not faring very well). 

Reviewer #3:

This paper combines modelling and experiments to investigate the relative success of chromosomal and plasmid-borne compensatory mutations for fitness costs of plasmid carriage. In general, there are a lot of things to like in this paper. In the modelling section, I found the general insights on the different roles of plasmid transfer for chrCM vs plaCM enlightening (i.e. reducing the fitness of your competitors through "weaponisation" vs direct spread of the CM). In the experimental work, especially the experiments showing RPS-like dynamics between plasmid-free, non-compensated, and plaCM cells and their direct link to model results stood out, as well as the results on the superiority of chrCM over plaCM in this system. These results have the potential to be insightful and relevant to a broad readership. However, I do have several important reservations towards the current manuscript that I think should be addressed.

One of the main results presented in the modelling section is that plaCM should succeed in its competition with plasmid-free cells and non-compensated carriers for a broader range of parameters than chrCM. I find the focus on this conclusion unfortunate for two reasons. First, as illustrated in Figure 1, this "broader range of parameters" is considered in terms of the (difference between the) conjugation rates of plasmids from uncompensated cells and from cells that have a CM. While for plaCM I can well imagine that there is a trade-off between fitness effect and transmissibility (as also found by Bethke et al, Dimitriu et al, and Turner et al; referred to on lines 168-169), for chrCM this is less obvious to me. In general, the fact that plaCM is present in the equilibrium for a broader range of the parameters considered does not say anything about the biological likelihood of these parameter conditions occurring (and hence of expecting to find these results). I would therefore ask the authors to comment on the biological relevance of the parameter regime considered (especially the broad range of gamma_c, and the comparable range of gamma_c and gamma_q), and to reconsider whether their conclusion on the expected success of plaCM vs chrCM is justified. 

Second, the focus on broad parameter ranges in my view hinders the comparison of model predictions and experimental data. All experiments are done on the same plasmid and CMs. Apart from the selection experiment, which does entail a change in eta, the experimental results hence basically hold for only one set of parameters (which the authors have actually measured!). The different experiments entail different initial conditions, but not changes in the parameters. Hence statements such as this one in the abstract "while plaCM was predicted to succeed under a broader range of parameters in mathematical models, experimentally chrCM dominated under all conditions" do not make sense to me. I think that throughout the manuscript, it should be made clearer what was varied in the different experiments (i.e. mostly who was initially present and in what fractions, and who was labeled) and how this relates to changes in the model (i.e. mostly changes in initial conditions, not parameter values). This may also include reconsidering what the main message of the modelling section should be, such that it connects best to the experimental results.

A stronger connection between modelling section & experiments can also be obtained by making the modelling section more accessible to all readers. The way it is written now, I think the results in the modelling section will be very hard to understand for the broad readership of PLOS Biology. Specifically, I think more care can (and should) be taken to avoid and/or better explain terminology such as "the expect stable fixed point of the underlying system" (line 139) and "a linear addition to the stability conditions" (line 153). Also, I personally always find it very helpful if some example dynamics are shown, because such plots are much easier to interpret than plots like Figure 1. Especially for the RPS-like dynamics (e.g. like shown in Fig 8), this may help readers understand the dynamics of the model. 

On the other hand, as a more mathematically inclined reader I was confused by some of the terminology used and I think this confusion can be avoided. In several places, the authors state that the 4-equation system cannot be solved analytically, while the 3-equation systems can. To me, and I would think to many mathematically trained readers, solving a system of ODEs analytically typically means that analytical expressions are found for f(t), p(t), c(t), etc. This is however not what is meant: the equilibria of the system of ODEs and their stability are determined analytically. I was similarly confused by the use of the term "phase portrait" for Figure 1, because in a phase portrait I would expect to see state variables on the axes (e.g. "p", "c"), not parameters. Lastly, I think more care should be taken in describing the results for the regime in which there is coexistence between f, p and q. It is not clear to me what the authors mean by a "linearised adaptive dynamics approach" (line 177 & SI), or why an AD approach could not pick up their result. If I understand correctly, in this parameter regime the "boundary" equilibria ((f, 0, 0), (0,p,0) and (0,0,q)) are unstable, right? Then I would think pairwise invasion analyses like the ones common in AD should pick up on the coexistence? Also, both in the main text and the SI it is claimed that this coexistence occurs in a stable fixed point (spiral). However, this could not be shown analytically. How can you be sure the coexistence does not occur on some other attractor, like a limit cycle?

In the model, spontaneous loss of the plasmid was not included, while this is sometimes done in these types of models. I do not think it is necessary to redo all the analyses including plasmid loss in the model, but I do think this assumption should be acknowledged and if possible, defended. Are spontaneous loss rates low for the specific plasmids considered? And if not, how would including spontaneous plasmid loss influence the results?

The experiments clearly show that for the specific system considered, chrCM is superior to plaCM. However, I am not fully convinced that the authors' claim that this is (partly) due to the weaponising of the plasmid (as is e.g. mentioned in the abstract) is justified. While the authors clearly show that this weaponisation can be important in the competition of chrCM vs plasmid free cells, in their particular system this effect may not play a major role in explaining the observed superiority of chrCM over plaCM because plaCM has low fitness anyway, and cannot fully spread in a susceptible population. Is the superiority of chrCM over plaCM then not just due to the difference in fitness (disregarding the weaponising effect)? I think the model offers a great opportunity to test this, namely by simulating the competition between plaCM and chrCM for gamma_c = 0. If chrCM cannot take over under those conditions, the conclusion can be drawn that the weaponising effect is necessary to explain the observed results.

One of the final conclusions of the paper is that, because chrCM is superior to plaCM, one may expect to find "hubs" of plasmid carriers because their chrCM allows for many plasmids. However, while in the system studied in this paper chrCM was indeed superior to plaCM, we are left wondering how general this result is and it would be good if the authors could critically discuss this. Again, the model may provide additional opportunity here: under what parameter conditions do you expect chrCM to be superior to plaCM? And should we expect these parameter conditions to be often met in nature?

I would like to end by stressing that I am not raising all these points to burn down the paper, but rather because the paper interested me and I would like to see the information in it become available for a broad audience. I hope these comments can contribute to doing that.

MINOR TEXTUAL COMMENTS / TYPOS

Lines 52-58 & 60-62 basically state the same thing twice -> consider reducing the text here?

Line 95: "3-equation ODEs" -> remove first "equation"?

Line 151: beta_q is mentioned here but this has not yet been defined.

Line 185, end: "though" -> "through"

Line 283: "previous worked" -> "work"

SI, Basic model: "grown rate" -> "growth rate"

SI, Plasmid compensation model, page 3: "This fixed point exists in the band ..." -> "This fixed point is positive in the band ..." (at least, my strong suspicion is that this fixed point also exists for other parameter values, just not in the positive quadrant.)

---

## [Decision Letter · Decision Letter 2]

2 Oct 2024

Dear Jamie,

Thank you for your patience while we considered your revised manuscript "Superiority of chromosomal compared to plasmid-encoded compensatory mutations" for publication as a Update Article at PLOS Biology. This revised version of your manuscript has been evaluated by the PLOS Biology editors, the Academic Editor and the original reviewers.

Based on the reviews, we are likely to accept this manuscript for publication, provided you satisfactorily address the remaining points raised by the reviewers and the following data and other policy-related requests.

IMPORTANT - please attend to the following:

a) Please make your Title more explicit by changing it to "Chromosomal mutations compensate for the costs of carrying a plasmid better than plasmid-encoded ones."

b) Please attend to the remaining concerns from the reviewers. Reviewer #1 now checks Accept and has no further concerns. Reviewer #2 simply has one textual request. Reviewer #3, however, has a few remaining points; one is a presentational query about Fig 1, and the other relates to a problem that s/he encountered when running your model (and might require further analysis).

c) Please address my Data Policy requests below; specifically, we need you to supply the numerical values underlying Figs 2, 3, 4, 5, 6, 7, S1ABC, S2, S3, S4, S5, S6, S7, S8, S9, S10, S11, either as a supplementary data file or as a permanent DOI’d deposition. I note that you already have data and/or code in depositions in Github, Liverpool DataCat and CEH, as well as in the zipped supplementary folder. There seems to be quite a lot of overlap between these, so it might be best to focus on one of them as place to direct readers for the data and code needed to recreate the Figures. It seems that Github might be the most comprehensive, as it contains the code and the Shiny app? However, because Github depositions can be readily changed or deleted, please make a permanent DOI’d copy (e.g. in Zenodo) and provide this URL (see below).

d) Please cite the location of the data clearly in all relevant main and supplementary Figure legends, e.g. “The data underlying this Figure can be found in S1 Data” or “The data underlying this Figure can be found in https://zenodo.org/records/XXXXXXXX

e) Please make any custom code available, either as a supplementary file or as part of your data (Zenodo?) deposition.

We expect to receive your revised manuscript within two weeks. 

*Published Peer Review History*

*Press*

Sincerely,

Roli

Roland Roberts, PhD

Senior Editor

rroberts@plos.org

PLOS Biology

DATA POLICY:

Regardless of the method selected, please ensure that you provide the individual numerical values that underlie the summary data displayed in the following figure panels as they are essential for readers to assess your analysis and to reproduce it: Figs 2, 3, 4, 5, 6, 7, S1ABC, S2, S3, S4, S5, S6, S7, S8, S9, S10, S11. NOTE: the numerical data provided should include all replicates AND the way in which the plotted mean and errors were derived (it should not present only the mean/average values).

CODE POLICY

DATA NOT SHOWN?

REVIEWERS' COMMENTS:

Reviewer #1:

[identifies himself as Alvaro San Millan]

I am happy with the modifications made by the authors.

Alvaro San Millan

Reviewer #2:

The authors have satisfactorily responded to my comments. In general I also thought the new version read better following the reshuffling based on other reviewers comments. 

I am still curious after the addition of lines 80-82 in the new version. If plaCM was the predominant CM in populations in which the plasmid persisted by infectious transfer - this suggests it was still more fit in the conditions of that given evolution experiment, despite this not being the case in the experiments presented here. Was there some difference in the design of both experiments that could suggest an explanation for this, which could be mentioned in the discussion? 

Reviewer #3:

The authors have addressed most of my comments very well, and I thank them for their efforts. The model section is now easier to read, and in general the results are well-explained. I also enjoyed the shiny-app, which I now explored a bit more. However, there are still two points that I think should be addressed before publication.

Point 1 (minor): Grey area in figure 1

a. I forgot to comment on this during the first round (for which I apologise), but it its unclear to me why the grey areas in Fig 1 (where gamma_C/Q > gamma_P) do not start from the origin, but rather from higher on the y-axis? Is the intersection of the axes not at gamma_C/Q = gamma_P = 0? If so, this should be indicated along the axes or in the legend. If I misunderstood and the intersection of the axes is actually gamma_C/Q = gamma_P = 0, I would recommend adjusting the text in the legend describing the grey area to avoid confusion.

b. In the legend of Fig 1, it is stated that in this grey area "the wild-type plasmid is always lost with the system reverting to the well known two-member-model of plasmid-free and plasmid containing type, as described in the Suppl Appendix." If I understand correctly, though, this is not entirely correct. For the plaCM model, yes, but for the chrCM the basic model in the suppl appendix does not hold as conjugation with chrCM cells as donor does not produce more chrCM cells, but rather plasmid-carrying, non-compensated cells. This is also the reason why you get bistability for any value of gamma, as is explained in the manuscript (lines 157-167). This point is easily remedied either by removing the grey areas from the figures in Fig 1 and showing the phase diagram for the whole space (which would have my preference for completeness), or by altering the text in the legend explaining why the gamma_C/Q > gamma_P condition was considered irrelevant.

Point 2 (somewhat more major): Inability of the model to replicate experimental results

This point arose from some simulations I did based on a misunderstanding of the point I rose earlier about whether the "weaponisation"-effect actually plays a role in the experimental system under study. While the authors clearly explain what would happen in a system with only plaCM and chrCM present, and in the "chrCM-only" model when gamma_c is equal to zero, I actually meant to suggest to test this condition in the full, simulated system (i.e. with both plaCM and chrCM present, as well as plasmid-free cells and non-compensated plasmid-carriers - so a "f + p + c + q"-model mimicking the experimental conditions of e.g. Fig 3). My hypothesis was that if one could replicate the experimental results (i.e. the superiority of chrCM) while gamma_c was equal to zero, this would suggest that the weaponisation was not important in this specific case, while if one could not find chrCM-superiority for gamma_c = 0 but this could be found for gamma_c > 0, this would indicate that the weaponisation effect is necessary to explain the experimental results. When I did some simulations* to try and test this myself, however, I ran into the problem that I was unable to replicate the experimental results of e.g. Fig 3 and 5 with the model, even when plugging in much lower conjugation rates for plaCM (100 fold lower; as was e.g. done in the paper for Fig S8 to mimic the RPS-dynamics found in Fig 7; or even 1000-fold lower). Maybe I didn't try hard enough or made a mistake - if so I would be happy to be corrected. If not, however, I think this is a problem, because in the discussion it is suggested that the difference between experimental results and model predictions can likely be explained by a yet-to-be-unveiled mechanism that would lower the effective conjugation rate of plaCM, but the simulation exercise suggests that something else is going on. I recognise that it is beyond the scope of the current work to do a full analysis of the f-p-c-q-model, but I do think that it should be acknowledged that as the model currently stands, it is not able to explain the observed superiority of chrCM even if plaCM would have much lower conjugation rate. I would therefore urge the authors to use their already existing simulation set-up to simulate the experiments of Fig 3 and Fig 5 with their measured parameters but much lower conjugation rate for plaCM (as they did in Fig S8 for the experimental results of Fig7) and then to comment in their discussion about any discrepancies between modelling predictions and the experimental results.

* What I did was simulate the f-p-c-q-model using the parameters in Table S1 (including the correction of conjugation rates for carrying capacities, as nicely explained in the table) and then simulate dynamics for different initial conditions (including ones with high values of c, to avoid missing c-dominated equilibria that are only reached above a certain manifold) to get a sense of the possible dynamics and equilibria. I could easily replicate the RPS-dynamics of Fig 7 and S8 by plugging in a 100-fold lower conjugation rate for q, but if I then took these parameters (with gamma_c = gamma_p) to simulate dynamics of the f-p-c-q-model I found that it still predicted fixation of q. When lowering q's conjugation rate even more (1000-fold), I found some large amplitude oscillations, but in all of the simulations I did this still ended with c disappearing from the system.

---

## [Editor Report · Decision Letter 3]

5 Nov 2024

Dear Jamie,

Thank you for the submission of your revised Update Article "A chromosomal mutation is superior to a plasmid-encoded mutation for plasmid fitness cost compensation" for publication in PLOS Biology. On behalf of my colleagues and the Academic Editor, Arjan de Visser, I'm pleased to say that we can in principle accept your manuscript for publication, provided you address any remaining formatting and reporting issues. These will be detailed in an email you should receive within 2-3 business days from our colleagues in the journal operations team; no action is required from you until then. Please note that we will not be able to formally accept your manuscript and schedule it for publication until you have completed any requested changes.

Sincerely, 

Roli

Senior Editor

PLOS Biology

rroberts@plos.org